# Behavior-specific occurrence patterns of Pinyon Jays (*Gymnorhinus cyanocephalus*) in three Great Basin study areas and significance for pinyon-juniper woodland management

**John D. Boone**[1]*, **Chris Witt**[2], **Elisabeth M. Ammon**[1]

**1** Great Basin Bird Observatory, Reno, Nevada, United States of America, **2** U.S. Forest Service Rocky Mountain Research Station, Boise, Idaho, United States of America

* boone@gbbo.org

**Data Availability Statement:** All relevant data are within the paper and its Supporting Information files.

## Abstract

The Pinyon Jay is a highly social, year-round inhabitant of pinyon-juniper and other coniferous woodlands in the western United States. Range-wide, Pinyon Jays have declined ~ 3–4% per year for at least the last half-century. Occurrence patterns and habitat use of Pinyon Jays have not been well characterized across much of the species' range, and obtaining this information is necessary for better understanding the causes of ongoing declines and determining useful conservation strategies. Additionally, it is important to better understand if and how targeted removal of pinyon-juniper woodland, a common and widespread vegetation management practice, affects Pinyon Jays. The goal of this study was to identify the characteristics of areas used by Pinyon Jays for several critical life history components in the Great Basin, which is home to nearly half of the species' global population, and to thereby facilitate the inclusion of Pinyon Jay conservation measures in the design of vegetation management projects. To accomplish this, we studied Pinyon Jays in three widely separated study areas using radio telemetry and direct observation and measured key attributes of their locations and a separate set of randomly-selected control sites using the U. S. Forest Service's Forest Inventory Analysis protocol. Data visualizations, principle components analysis, and logistic regressions of the resulting data indicated that Pinyon Jays used a distinct subset of available pinyon-juniper woodland habitat, and further suggested that Pinyon Jays used different but overlapping habitats for seed caching, foraging, and nesting. Caching was concentrated in low-elevation, relatively flat areas with low tree cover; foraging occurred at slightly higher elevations with generally moderate but variable tree cover; and nesting was concentrated in slightly higher areas with high tree and vegetation cover. All three of these Pinyon Jay behavior types were highly concentrated within the lower-elevation band of pinyon-juniper woodland close to the woodland-shrubland ecotone. Woodland removal projects in the Great Basin are often concentrated in these same areas, so it is potentially important to incorporate conservation measures informed by Pinyon Jay occurrence patterns into existing woodland management paradigms, protocols, and practices.

**Funding:** Pinyon Jay studies in Eastern Nevada were funded by cooperative agreement # 2004-0171-000 between National Fish and Wildlife Foundation (www.nfwf.org) and Great Basin Bird Observatory (JB and EA). This funder had no role in study design, data collection, analysis, decision to publish, or preparation of the manuscript. Pinyon Jay work in Southern Idaho was funded by cooperative agreement # 12-JV-11221638-130 between U.S. Forest Service (www.fs.usda.gov) and Great Basin Bird Observatory (JB and EA). A representative of this funder is a co-author (CW) who contributed to study design, data collection, data analysis, decision to publish, and preparation of the manuscript. Pinyon Jay work in Central Nevada was funded by cooperative agreement # FAA080097 between U.S. Bureau of Land Management (www.blm.gov) and Great Basin Bird Observatory (JB and EA). This funder had a collaborative role in study design, but no role in data collection, analysis, decision to publish, or preparation of the manuscript. Collection of Forest Inventory Analysis data relevant to all study areas was supported internally by the U.S. Forest Service (see above).

**Competing interests:** The authors have declared that no competing interests exist.

# Introduction

The Pinyon Jay (*Gymnorhinus cyanocephalus*) is a highly social corvid that inhabits pinyon-juniper and other coniferous woodlands in the interior western United States [1–3] (Fig 1). Pinyon Jays form year-round flocks that can range from a few dozens to several hundred members [3–6]. They are perhaps best known for harvesting and caching the seeds, or "pine nuts", of the pinyon pine (primarily *Pinus monophylla* and *P. edulis*) as their primary food source, though they also consume other conifer seeds and insects [3, 7, 8]. Pinyon Jays occur in parts of at least ten states, but most of their range lies within Bird Conservation Regions (BCRs) 9 ("Great Basin") and 16 ("Southern Rockies and California Plateau") [9], with highest densities in east-central Nevada and western New Mexico (Fig 2). Since North American Breeding Bird Survey data collection began in 1967, Pinyon Jays have experienced steep and sustained declines averaging 3–4% per year both range-wide and within most of the states and regions they occupy [4, 10]. This equates to a loss of about 85% of the population over 50 years, one of the largest recorded declines among all widely-distributed passerine birds in the western United States over the same time period [10, 11].

Despite this decline, no systematic conservation efforts have been undertaken for the Pinyon Jay, although it is included on the 'sensitive species' lists of many federal and state management agencies and avian conservation organizations [12–15], and is the subject of a recent interagency working group conservation strategy [15]. The lack of conservation action for Pinyon Jays may be attributable to several factors. First, despite a rich knowledge of the species' social behavior, breeding behavior, and spatial memory [6, 16, 17], its occurrence and habitat use patterns are still poorly characterized in most parts of its range. Second, there are no widely accepted or strongly supported hypotheses about the causes of Pinyon Jay population declines. Finally, the landscapes inhabited by Pinyon Jays are primarily managed for other priorities. These include improving habitat for game species such as Greater Sage-Grouse (*Centrocercus urophasianus*) and mule deer (*Odocoileus heminus*), creating wildlife corridors, and mitigating fire hazards [3, 18–22].

Pinyon Jay declines could be related, at least in part, to changes in the pinyon-juniper woodlands (hereafter used to indicate woodlands that contain either pinyon pines, junipers, or both) that comprise most of their habitat [3, 22–24] and much of the forested landscape of the Great Basin (i.e., BCR 9) and Colorado Plateau (i.e., BCR 16) [25–28]. The spatial extent of pinyon-juniper woodlands (commonly a *Pinus monophylla—Juniperus osteosperma* association) in this region has undergone climate-induced fluctuations since the end of the Pleistocene epoch 11,500 years ago [29, 30], but it has been suggested by some authors that over the last 150 years, extension of local woodland range (i.e. "expansion") and increased tree densities within extant stands (i.e. "infill") have occurred at atypical and perhaps unprecedented rates, at least in the Great Basin [19, 22, 31–33]. Other authors have questioned this conclusion and suggested that expansion and infill are either localized, part of a historically normal pattern of spatio-temporal woodland dynamics, or recoveries from earlier widespread clearing during the western settlement period [22, 34, 35]. Alterations of fire regimes and land use patterns associated with western settlement, along with the impacts of climate change, further complicate understanding the dynamics of pinyon-juniper woodland change [22].

In the Great Basin, a primary pinyon-juniper woodland management objective over the last 20 years has been creation or restoration of shrubland habitat by clear cutting stands that are regarded as encroaching into shrublands, often with a goal of benefitting Greater Sage-Grouse [36–39]. If and how this type of vegetation management affects Pinyon Jays remains undetermined, as does the potential of these treatments to benefit Pinyon Jays if suitably designed. Answering these questions requires a better understanding of Pinyon Jay occurrence patterns

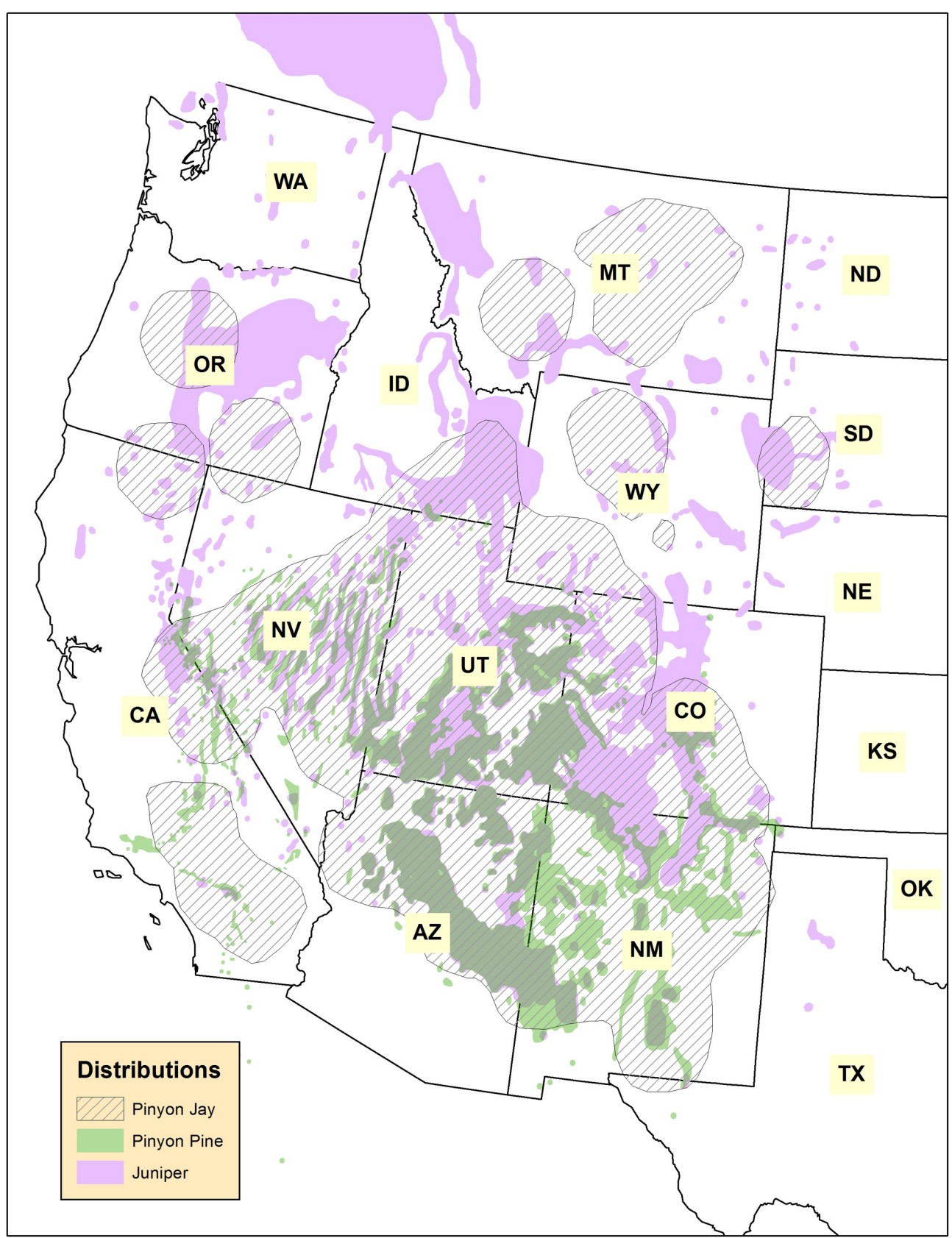

**Fig 1. Species distributions.** Distribution of the Pinyon Jay [4], pinyon pine (*Pinus monophylla* and *P. edulis* combined), and juniper (*Juniperus osteosperma*, *J. occidentalis*, and *J. spoculorum* combined) [5] in the western United States. State name abbreviations are shown. Areas that contain both pinyon pine and juniper are a darker color than pinyon pine or juniper alone.

and habitat use in the Great Basin, determining the extent of overlap between their preferred habitats and ongoing vegetation management activities, and monitoring or modeling the effects of those activities.

In this study, we used observational techniques to compare the habitats used by Pinyon Jays in the Great Basin for caching or retrieving seeds, for foraging, and for nesting, to the full range of habitat available within pinyon-juniper woodlands. Our goals were to determine whether Pinyon Jays in the Great Basin used predictable and measurable subsets of available habitat for these distinct behavior types. If so, this information could assist managers seeking to incorporate Pinyon Jay conservation measures into existing vegetation management programs for pinyon-juniper woodlands, suggest additional conservation actions, and help to guide future research [15].

## Methods

### General approach

We used direct observation and radio telemetry to record locations where Pinyon Jay flocks occurred in the Great Basin, and compared these locations to a set of pre-existing Forest Inventory and Analysis (FIA) plots established by the U.S. Forest Service (USFS) in pinyon-juniper woodlands that served as controls (see below for details). Pinyon Jay locations were recorded from 2008–2013 in three study areas (Fig 3). Control sites were distributed over a broader region located entirely within the Central Basin and Range Ecoregion [40] and BCR 9 (Fig 2) that encompassed all three Pinyon Jay study areas (Fig 3). They therefore provided a representative characterization of the diversity of pinyon-juniper woodland habitat available within the general study region. Habitat at both the Pinyon Jay locations and the control sites was quantified using the standardized FIA protocol (see below) within circular 0.405-ha (1-acre) plots, which defined the spatial scale of this analysis. Given the scattered distributional pattern of Pinyon Jay flocks, control sites were assumed to be unoccupied by Pinyon Jays, though they ostensibly could be. This approach best corresponds to a case-control study design [41, 42]. We recognize the potential bias inherent in case-control sampling designs and followed Keating and Cherry's [42] and Manly et al.'s [43] guidance for analysis. Analyses consisted of ordination and logistic regression, supplemented by data visualizations.

### Pinyon Jay study areas and timeline

Pinyon Jay locational records were collected from multiple Pinyon Jay flocks in these three study areas (Fig 3):

1. Eastern Nevada (~ 63,130-ha study area), specifically the foothills between Baker, Nevada and Great Basin National Park, and Steptoe Valley south of the town of Ely, Nevada. This area is mostly comprised of public lands managed by the U.S. Bureau of Land Management but includes some private property near Baker. Pinyon Jay data were collected from 2/28/2008–6/26/2008 and from 5/10/2009–8/27/2009.

2. Southern Idaho (~ 23,470-ha study area), specifically the area in and around City of Rocks National Reserve and Castle Rocks State Park in Cassia County. This study area contains the northernmost occurrence of pinyon pine in North America [44]. Jurisdictions within

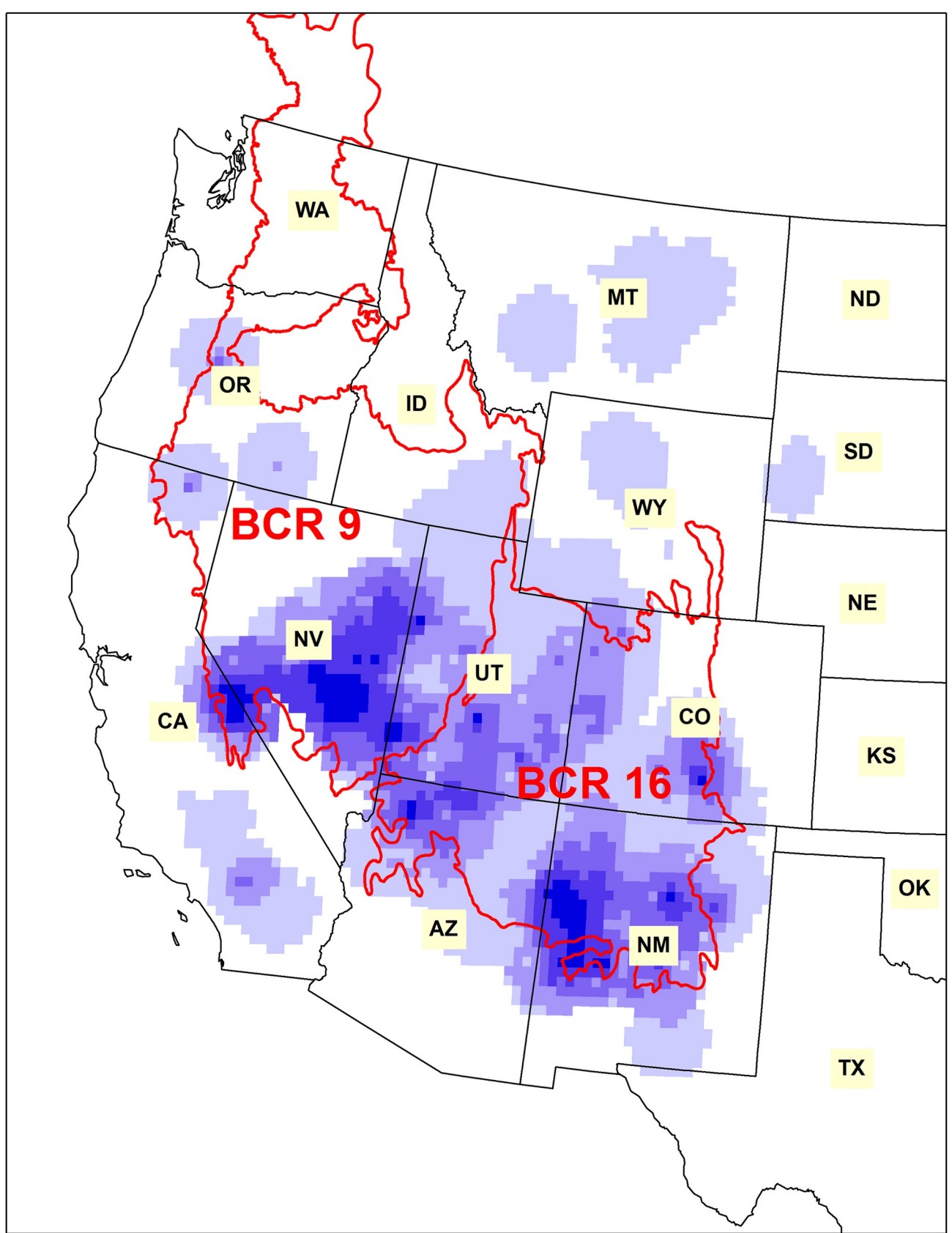

**Fig 2. Pinyon Jay density.** Relative density of the Pinyon Jay (purple colors, with darker colors representing higher relative densities [4]), also showing Bird Conservation Regions (BCRs) 9 (Great Basin) and 16 (Southern Rockies / Colorado Plateau) in the western United States. State name abbreviations are shown.

the study area are the U.S. National Park Service, Idaho State Parks, U.S. Bureau of Land Management, and private lands. Pinyon Jay data were collected from 7/20/2012–10/5/2012.

3. Central Nevada (~ 89,030-ha study area), specifically the Desatoya Range which lies along the border between Churchill County on the west and Lander County on east. This study area is comprised almost exclusively of U.S. Bureau of Land Management lands, with limited private inholdings. Pinyon Jay data were collected from 3/29/2013–6/5/2013.

Geographical coordinates of multiple Pinyon Jay locations within each study area are provided in S1 Table. Study area locations and periods of data collection were governed by three

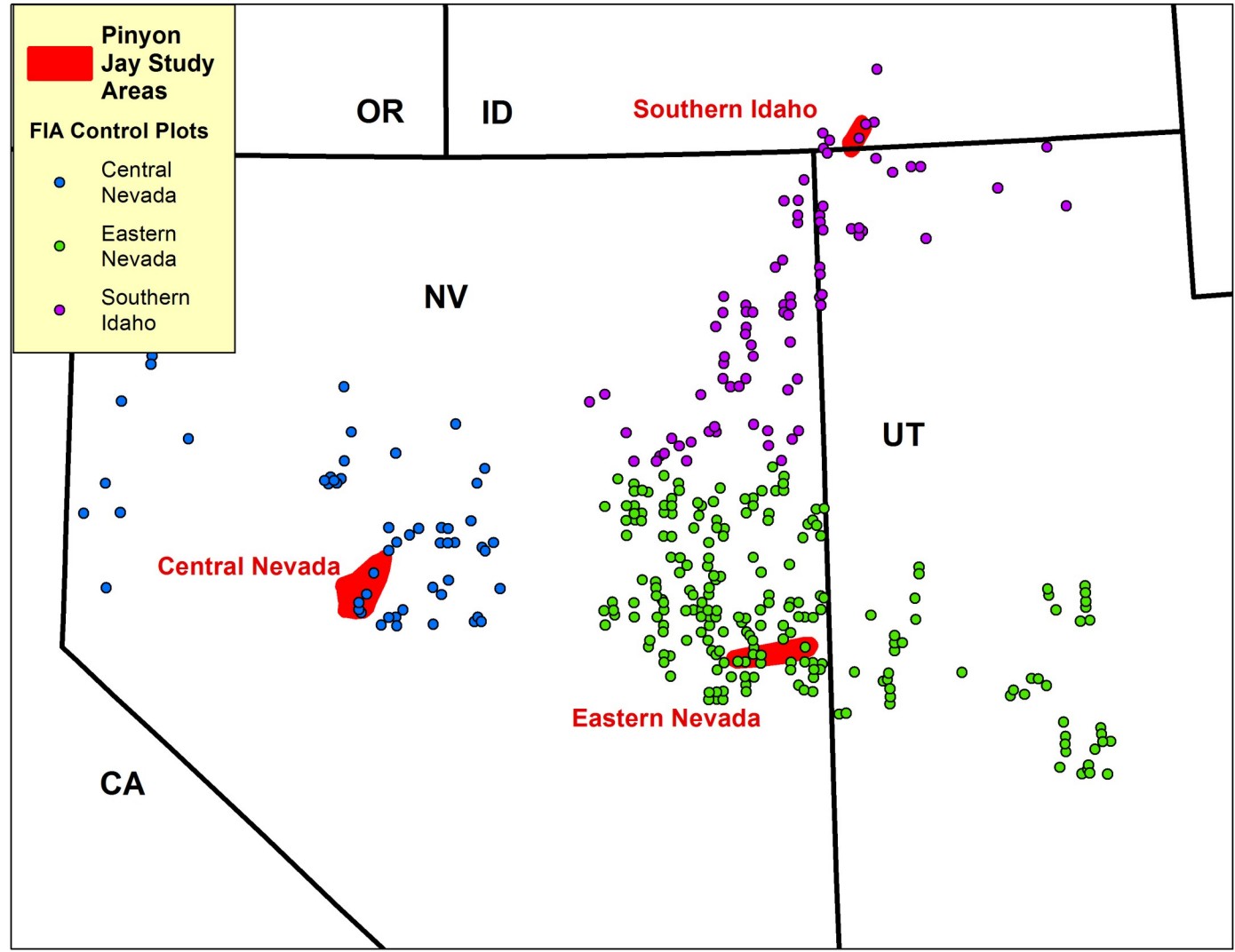

**Fig 3. Study areas and control sites.** Locations of three Pinyon Jay study areas and FIA plots that served as control sites in Nevada, Idaho, and Utah. FIA plot locations are approximate (i.e. "fuzzed") to comply with USFS data protection policies. FIA plots are color coded to indicate the Pinyon Jay study area to which they were assigned. Geographical coordinates of multiple Pinyon Jay locations within each study area and control sites are provided in S1 Table.

different funding agreements, and therefore seasonality was not standardized across all three areas. All study areas are characterized by mountainous "basin and range" topography, with plant communities dominated by big sagebrush (*Artemisia tridentata*) at lower elevations, pinyon-juniper woodlands at mid-elevations, and various mountain shrub and forest types at high elevations. Pinyon-juniper woodlands range in elevation from ~ 1,500 m– 2,600 m across all study areas and are usually comprised of varying proportions of *P. monophylla* and *J. osteosperma*. Public lands in all study areas are managed for multiple uses and experience varying levels of livestock grazing and off-road vehicle travel.

## Pinyon Jay data collection and processing

Pinyon Jay data collection had several distinct components; initial searches, observational surveys, capture and radio-tagging, and radio-telemetry surveys. The observational surveys and radio-telemetry surveys generated the Pinyon Jay locations that are the basis of our analyses.

Initial searches of potential Pinyon Jay habitat (defined as pinyon-juniper woodlands and visually adjacent shrublands) were conducted during the first 1–2 weeks of field work at a given study area on foot and by vehicle to identify all or most of the Pinyon Jay flocks present. Because Pinyon Jay flocks are visually apparent from long distances, "noisy" (except when at the nest), and spatially segregated from one another, we regarded this a feasible goal. Search patterns used during initial searches were not systematized but were instead tailored to take advantage of local topography and access points. Areas searched were marked on imagery maps to facilitate and ensure thorough coverage of pinyon-juniper woodland across a given study area. Upon detecting a flock, the observer usually maintained contact over a period of 1–3 h on each of several visits to obtain a preliminary and approximate understanding of the flock's movement patterns and primary activity area(s).

Flocks that were consistently detected during initial searches were then subjected to more intensive study by observational surveys and/or radio-telemetry surveys to obtain a sample of occupied locations. Observational surveys involved establishing visual contact with a flock; observing the flock with binoculars from a distance sufficient to prevent alteration of flock behavior (typically > 75 m); and recording once per hour (as nearly as practicable) the point coordinates of the estimated centroid of the flock's location along with estimated flock size and predominate behavior type (Table 1). The goal of observational surveys was to obtain locations across an entire daylight cycle at least once per week (usually assembled from several

**Table 1. Behavior types.**

| Behavior Type | Description |
|---|---|
| Caching | Birds observed either caching or retrieving previously cached pine nuts or other similar food items. |
| Foraging | Birds observed collecting any type of new food item in trees, shrubs, or on the ground, including pine nuts, other plant material, or insects. |
| Nesting | Nesting confirmed by one or more of the following: nest finding, direct observation of birds on nest, or observation of confirmatory breeding behaviors, such as carrying nest materials. |
| Roosting | Birds present during period(s) of darkness. |
| Loafing[1] | Birds observed resting, usually in trees, while not engaged in any other listed behaviors. |
| Flyover[1] | Entire flock flying in a directional manner, without landing. Short flight movements by one or a few birds while the flock was engaged in one of the other listed behaviors were not considered flyovers. |

Description of Pinyon Jay behavior types that were recorded during observational and radio-telemetry surveys.

[1]Because these behavior types could occur when flocks were in transit, they were excluded from all analyses and data visualizations.

observation sessions conducted during different time periods on different days) for each flock over the duration of the data collection period. Coordinates of flocks were usually obtained by recording observer position with a GPS unit; recording a bearing to the visually-estimated flock centroid with a compass; measuring distance to the flock centroid by rangefinder or estimating distance by eye (for shorter distances < 25 m); and then plotting the flock's estimated point location in GIS based on these parameters. In some cases, coordinates could be obtained directly by GPS after a flock vacated a previously occupied location. As observers became increasingly familiar with the daily movement patterns of a flock, which tended to be consistent, their efforts were increasingly focused on the portions of the daily activity cycle that were more difficult to characterize. This resulted in final data sets for each flock that represented all portions of the diurnal spectrum.

Radio-telemetry surveys were also used to obtain Pinyon Jay locations for some of the studied flocks. Two methods were used to capture Pinyon Jays for radio-tagging; baited walk-in traps and mist nets. Traps were used where Pinyon Jays could be consistently drawn (determined by camera traps) to a supplemental food station baited with shelled peanuts, sunflower seeds, and dried corn. In these areas, a home-made wood and mesh walk-in trap (0.6 x 1.2x 0.9 m) with an open door was placed at the site, and bait spread periodically inside the trap to habituate birds to entering the trap. When habituation was sufficient (determined by camera traps), capture attempts were made by rigging the door for remote manual release, observing activity at the trap from a blind ~ 25 m from the trap, and releasing the door with a pull cord when Pinyon Jays were inside the trap. Mist-nets were used in areas where walk-in traps were not viable, in locations routinely visited by Pinyon Jays where nets could be deployed without being easily detected by birds. Mist nets (60 mm mesh) were arrayed either singly, as doubles, or stacked, depending on the geometry of the woodland opening where they were erected. In some cases, call playback was also used to try to draw birds into the nets. Any Pinyon Jays captured by walk-in trap or mist-net were weighed and aged; standard aluminum leg bands (U.S. Geological Survey) were attached; and radio transmitters (Advanced Telemetry Systems model A2450) were glued onto feather stubble clipped to ~ 0.3 cm above skin level in the interscapular area. No more than six individuals from any single flock were radio-tagged, since data were collected to characterize flocks rather than individual birds. Captured birds were handled and processed only by experienced individuals holding a U.S. Fish and Wildlife Service master banding permit. Radio-tagged birds were manually tracked using a handheld three-element Yagi antenna with an Advanced Telemetry Systems R410 or R100 receiver.

The goal of radio-telemetry surveys was to collect hourly locations over one entire daylight cycle per week for each flock, either in a single long session or over several sessions of 3–5 hours on different days and at different times. Most often, telemetry fixes from one or more radio-tagged flock members were used to approach the flock and establish visual contact, at which point locational and behavior type attributes were collected exactly as described above for observational surveys. Occasionally, flock locations were estimated by biangulation or triangulation of telemetry bearings that were post-processed using LOAS software (Ecological Software Solutions, LLC). In cases where the flock was not directly observed, behavior type was only recorded when it could be reasonably inferred based on previous observations (i.e. return to a nest colony location) or context (roosting locations at or after sunset).

For a given flock, locations could be obtained by either observational surveys, radio-telemetry surveys, or both, but because both approaches ultimately relied on the same observational process to estimate location and determine behavior type, we regarded the resulting data as comparable. Initially, radio-telemetry was the preferred survey approach, but by later data collection periods we determined that visual contact with most flocks could routinely be

established without reliance on radio-tagged flock members, and our efforts to capture birds for radio-telemetry were discontinued.

The full set of Pinyon Jay locations recorded during field work were filtered and processed prior to analysis. First, any locations where no behavior type was recorded were deleted. Next, all "flyover" locations and loafing locations (Table 1) were deleted, based on the premise that these could occur throughout and sometimes beyond the flock's core home range. Finally, to avoid over-leveraging data from locations where flocks were easiest to observe, any behavior-ally specific records that were closer together than 71.8 m were spatially averaged using the Mean Center tool in ArcMap 10.5 (ESRI, Redlands, CA) to generate a single location. This was the smallest threshold distance sufficient to prevent any overlap between the 0.405-ha habitat plots used to assess habitat (see below), so this process essentially smoothed Pinyon Jay loca-tions to a spatial scale that matched the habitat assessment scale. After these steps, 154 Pinyon Jay locations with recorded behaviors were retained for analysis (see S1 Table).

### Control site selection

Control sites were selected from the pre-existing FIA plots visited and measured between 2005–2013. FIA data provide a probabilistic and geographically unbiased assessment of forest / woodland attributes over time and space [45, 46] and a robust dataset to describe habitat for multiple species [47–51]. One FIA plot is randomly positioned within every cell of a sampling grid that covers all public and private forest land in the United States, with an overall density of one plot per 2,428 ha (6,000 acres) within the extent of that sampling frame [52]. Each year, 10% of these plots are surveyed or re-surveyed in the western U.S. using a spatially-interpene-trating sampling design that avoids conflation of spatial with temporal trends [45]. The criteria used to select the subset of FIA plots that were appropriate control sites for this study were as follows:

1. The FIA plot had to be < 200 km from the nearest Pinyon Jay location retained for analysis.

2. Presence of pinyon-juniper woodland within the FIA plot had to be confirmed by direct observation of the field assessment crew during the most recent assessment visit. A wood-land type classification based solely on remote-sensing data was not sufficient to meet this criterion.

3. The site had to be located in both the Central Great Basin Ecoregion and in BCR 9.

4. The most recent assessment visit had to have occurred in 2005 or later.

In total, 346 FIA plots from the larger data set met these criteria (Fig 3) and were included in our analysis (S1 Table gives geographical coordinates for each control site). Each control site was assigned a regional attribute that corresponded to the closest Pinyon Jay study area (n = 212 for the Eastern Nevada region, n = 81 for the Southern Idaho region, and n = 53 for the Central Nevada region). Control sites were not distributed symmetrically around the Pin-yon Jay study areas (especially for Southern Idaho) because their extent was constrained by the distribution of pinyon-juniper woodland and by the original USFS site selection process.

### Habitat assessment

Woodland habitat at Pinyon Jay locations was characterized using current FIA field protocols [53]. Site attribute measurements at Pinyon Jay locations on non-woodland locations followed procedures outlined for a FIA All-Conditions Inventory [54]. For all Pinyon Jay activity sites, FIA plot centers were placed at the Pinyon Jay location point coordinates and assessments were performed by USFS crews fully trained in FIA protocols and procedures. Control sites

were assessed as part of routine USFS operations in various years between 2005–2013 [55–57]. Pinyon Jay locations were assessed during the 2013 (Eastern Nevada and Southern Idaho study areas) and 2015 (Central Nevada) FIA field seasons. Assessments of all Pinyon Jay locations and nearly all control sites occurred within the seasonal window (April–November) specified by the FIA protocol, with a small number of control sites (9 of 346) assessed just outside this window (S1 Table).

FIA plot layout is based on a 0.405 ha circle that defines four subplots of 7.3 m diameter within which actual measurements are made. One subplot is located at the center of the larger circle, and the other three subplots have their centers equally spaced along its circumference. On a standard forested FIA plot, over 120 attributes are measured within each subplot to characterize location, condition, and vegetation [58]. Subplot data are then averaged or summed over subplots as appropriate and extrapolated to generate data at the whole-plot scale. The subset of FIA attributes that were considered for use in this study are presented and briefly described in Table 2.

To help interpret analytical results, we decided to create one additional non-FIA attribute, "Distance to Edge" (Table 2) which is the shortest linear distance between a Pinyon Jay location or control site and the lower-elevation woodland-shrubland ecotone. First, polylines were digitized in ArcMap 10.5 to delineate the approximate ecotonal boundary based on visual examination of imagery. Then, the shortest distance from each Pinyon Jay location or control site to the polyline was computed using the Near tool in ArcMap 10.5. This value was usually positive but could be negative if a Pinyon Jay location was in shrubland at a lower elevation than the polyline. Because the Distance to Edge metric was not a FIA attribute, it was used only for data visualizations, not for statistical analyses.

## Ethics and permissions

Data collection procedures were primarily observational and were not submitted to or approved by an Institutional Animal Care and Use Committee. Field work that involved capture, radio-tagging, and release of Pinyon Jays was conducted under U.S. Department of the Interior Federal Bird Banding Permit # 22912 to Elisabeth Ammon, Idaho Fish and Game Department Collection Permit # 120724, and Nevada

Department of Wildlife Collecting Permit # 29948. These procedures are typical of field work involving bird capture and banding that do not involve any prolonged handling, confinement, or euthanasia, and where birds are released unharmed within a few minutes of being captured and tagged. Wallace Keck, Superintendent of City of Rocks National Reserve and Park Manager for Castle Rock State Park, allowed use of park facilities, with further assistance from Trenton Durfee. Many local landowners permitted us to access to their property, notable among them LeAnn and Kim Draper, whose property functioned as a capture location and staging area for telemetry efforts.

## Analysis

**Data.** The complete data set used for all analyses and data visualizations is provided in S1 Table. Roosting locations were omitted from all analyses listed below due to small sample size (n = 3).

**Ordination.** To visualize how the habitat characteristics of Pinyon Jay locations for each behavior type overlapped with control sites, we performed a principle components analysis (PCA) on continuous FIA habitat attributes (Table 2) using the princomp function in base R (v3.5.1; [60]). All ordinated attributes were standardized with a z-transformation across all regions to promote optimization.

**Table 2. Habitat attributes.**

| Attribute | Ordination | Logistic Regression | Data Visualization |
|---|---|---|---|
| **FIA Attributes** | | | |
| Elevation (measured in feet at plot center) | Y | Y | Y |
| Slope (expressed as slope percentage, or {{rise/run} x 100} at plot center) | Y | Y | Y |
| Habitat Type [59] | NA | N[1] | NA |
| Stand Age (age in years of the oldest pinyon pine or juniper tree on plot, as determined by coring) | Y | Y | Y |
| Stand Density Index (index of three-dimensional tree density within stand [59]) | N[2] | N[2] | Y[2] |
| Canopy Cover (% by line transect) | Y | N[3] | N[3] |
| Tree Cover (% by ocular estimation) | Y | Y | Y |
| Shrub Cover (% by ocular estimation) | Y | Y | Y |
| Forb Cover (% by ocular estimation) | Y | Y | Y |
| Grass Cover (% by ocular estimation) | Y | Y | Y |
| Woody Debris (count of pieces of dead woody debris material along transects for seven different size diameter classes, defined by twig / branch diameter ranges) | Y | Y[4] | Y[4] |
| Distance to Road (km from plot center, assigned using topographic maps; original ordinal ranges converted to range midpoints) | Y | Y | Y |
| Disturbance Type (categories of silvicultural treatment or other disturbance occurring within the previous five years) | NA | N[5] | NA |
| Disturbance Presence ("yes-no", for any disturbances within the last five years) | NA | Y | NA |
| **Non-FIA Attribute** | | | |
| Distance to Edge (Distance to the lower-elevation woodland-shrubland ecotone, measured directly in on imagery in ArcMap 10.5) | N[6] | N[6] | Y |

Attributes describing Pinyon Jay habitat that were considered in this study. Brief descriptions of attributes are provided parenthetically, with complete descriptions of associated methodologies available in [59]. The 2[nd], 3[rd], and 4[th] columns indicate whether or not an attribute was used (Y = yes, N = No, NA = categorical attribute, not applicable) for the ordinations, logistic regressions, and data visualizations that are presented below. Footnotes provide additional explanations about attribute use considerations.

[1] All control sites used in analysis were characterized by presence of at least some pinyon pine and/or juniper trees, but the FIA habitat type distinctions were too fine-grained (33 different types, median number of sites / type = 5) for inclusion in analysis.

[2] Not considered in analyses because this was an index derived from more fundamental FIA attributes, but included in data visualizations because of possible interpretational value.

[3] Omitted because of high correlation to Tree Cover attribute (r = 0.68).

[4] Reduced to a single attribute for logistic regression and data visualization using PCA that described 86.1% of variation across all original classes.

[5] The level of articulation was too fine for the limited number (n = 68) of disturbed sites and too unevenly distributed among types.

[6] Not considered for ordination or logistic regression analysis as a non-FIA attribute, but used for data visualizations for possible interpretive value

**Logistic regression.** To evaluate the effects of measured habitat attributes on Pinyon Jay occurrence patterns by behavior type [41, 43, 61] we performed separate logistic regressions for caching, foraging, nesting locations using a generalized (binomial) mixed model fit with the glmer function in the lme4 package (v1.1–19; [62]) in R (v3.5.1; [60]). Within each behavior-specific logistic regression model, region was treated as a random effect and selected FIA habitat attributes (see next paragraph and Table 2) were treated as fixed effects. Because the ratio of Pinyon Jay locations to control sites was fairly low, the odds ratio output from logistic regression can be treated as an approximation to the resource selection function [41, 63, 64]. However, it is important to recognize that the intercept value of the logistic regression does not estimate the overall use probability, as the Pinyon Jay locations were not randomly selected [41].

Initial evaluation of available predictor attributes (Table 2) consisted of plotting data pairs, evaluating correlations, and preliminary overall model fitting to ensure base-level

convergence. Several attributes were eliminated from the analysis due to high correlations with other attributes or highly uneven distribution of values, and others were combined into a single attribute with PCA (Table 2). All continuous attributes used for analysis (Table 2) were converted to metric units and z-transformed to facilitate model fitting and term comparison.

Rather than using the same set of control sites for each of the three logistic regressions, which would inflate Type I error, we split the control sites among the three behavior-specific models. Splitting was randomized and permutated 10,000 times to account for variation in the splitting process. An advantage of this approach was that it allowed information from all control sites to inform each behavior-specific model once all permutations were combined (see below) without inflating Type I error. For each permutation, the control sites were split on a region-by-region basis among behavior types, in proportion to the frequency of each behavior type within a given region. Then the control sites assigned to a given behavior type were combined across the three regions and analyzed.

The results from multiple model permutations were then combined to estimate overall terms. This was done by first discarding any specific permutations that did not converge due to unreasonable splits of control sites that could occur occasionally within a specific allocation permutation. Then, for each retained permutation, a single set of fixed effects parameters were drawn from the multivariate normal distribution described by the glmer model fit using the rmvnorm function in the mvtnorm package (v1.0–8; [65]) in R (v3.5.1; [60]). By drawing from the distribution within each permutation, the full set of parameter values across permutations includes both parameter estimation uncertainty and control allocation uncertainty. To avoid distributional assumptions in evaluating the significance of the parameter estimates, we tested whether the resulting estimates significantly differed from 0 by using a permutation-style two-tailed approach [66] with an empirical cumulative distribution function (calculated using the ecdf function in the stats package in base R (v3.5.1; [60]) to estimate the two tail probabilities with respect to 0, taking the smaller value as the focal tail, and doubling that tail's probability. We combined the estimates of the random effects across all of the permutations to generate the distribution of values for each random effects term across the uncertainty in control site allocation.

To evaluate the overall models, within-sample classification accuracy for each Pinyon Jay behavior type was averaged over all retained permutations. Because the number of Pinyon Jay locations for each behavior type was limited, we did not withhold a subset of locations for external validation.

**Data visualization.** To aid in the interpreting statistical results and to highlight univariate patterns of interest, box plots were created for most of the continuous attributes shown in Table 2 to compare the distribution of attribute values for behavior-specific Pinyon Jay locations and control sites. Box plots were also created for two attributes not used in analyses–the stand density index and distance to edge (Table 2)–because of their potential to provide useful indicator metrics encompassing the patterns seen in the ordination and logistic regressions.

## Results

### Data collected

Behavior-specific Pinyon Jay locations (n = 154) were obtained from 15 different flocks. Details about distribution of Pinyon Jay behavior type locations among study regions, number of flocks per study area, methods of data collection used within study areas, and allocation of control sites among behaviors and regions for the logistic regression analyses are summarized in Table 3. Pinyon Jay data were not fully symmetric among the study areas and behaviors. More specifically, nesting was not recorded in Southern Idaho because the data collection

**Table 3. Summary of Pinyon Jay locations and control sites.**

| Study Area / Region | Data Type | Caching | Foraging | Nesting | Roosting[1] | Total | # Flocks [Method of Study] |
|---|---|---|---|---|---|---|---|
| Eastern Nevada | Pinyon Jay Location | 12 | 0 | 12 | 3 | 27 | 2 flocks [Flock #1 = T(6); Flock #2 = T(6)] |
| | Allocated Control Sites for LR | 106 | 0 | 106 | N/A | 212 | |
| Southern Idaho | Pinyon Jay Location | 32 | 19 | 0 | 0 | 51 | 2 flocks [Flock #1 = O & T(6); Flock #2 = O & T(2)] |
| | Allocated Control Sites for LR | 51 | 30 | 0 | N/A | 81 | |
| Central Nevada | Pinyon Jay Location | 26 | 22 | 28 | 0 | 76 | 11 flocks [All = O] |
| | Allocated Control Sites for LR | 18 | 15 | 20 | N/A | 53 | |
| Total (All Regions) | Pinyon Jay Location | 70 | 41 | 40 | 3 | 154 | |
| Total (All Regions) | Total Control Sites | 175 | 45 | 126 | N/A | 346 | |

Summary of Pinyon Jay locations retained for analysis after data processing, number of control sites, number of flocks studied by study area, and study methods by flock. Pinyon Jay locations are broken down by region and behavior type. Control sites are broken down by their regional attribution and by their allocation to each behavior type in the logistic regression models. The final column shows the number of different Pinyon Jay flocks from which data were collected in each study area, with the data collection methods used shown in brackets (T = telemetry surveys, with number of deployed radio tags in parentheses; O = observational surveys). N/A = not applicable. LR = logistic regression.

[1]Roosting locations were omitted from all analyses and visualizations due to small sample size.

period excluded the breeding season, and our data recording protocol in 2008–2009 for Eastern Nevada did not specify recording the foraging behavior type. Only caching locations were recorded in all three study areas, and only in the Central Nevada study area were all three major behavior types (caching, foraging, and nesting) recorded (Table 3).

Every Pinyon Jay location and control site is shown in S1 Table, along with associated habitat attributes.

## Ordination

The ordination reduced the input habitat attributes (Table 2) to a two-dimensional representation that explained 36.5% of the total variance in the data set. Pinyon Jay locations tended to be spatially offset from control sites in this two-dimensional space, and Pinyon Jay caching, foraging, and nesting locations formed distinct but overlapping clusters, with caching locations being the most segregated (Fig 4). The first component (21.2% of total variance) described a gradient ranging from: a) lower-elevation, flatter areas with younger, more open woodlands with a significant shrub, forb and grass component (i.e. negative values on the X-axis of Fig 4), to b) higher-elevation, steeper areas with older, denser woodlands and relatively little shrub, grass, and forb cover (i.e. positive values on the X-axis) (Table 4). Along this axis, Pinyon Jay caching locations had the lowest values on average, followed by foraging locations, nesting locations, and control sites (Fig 4). The second component (15.3% of total variation) described a gradient from denser woodland with more fine woody debris (i.e. negative values on the Y-axis) to more open woodlands with less fine woody debris (i.e. positive values on the Y-axis) (Table 4). This gradient was independent of elevation (Table 4). Along this axis, there was considerable overlap between control sites and Pinyon Jay locations. However, different Pinyon Jay behaviors tended to segregate from one another along this axis (albeit with substantial overlap), with caching locations having the highest values and nesting locations the lowest. A plausible biological interpretation for these results is that the first component describes larger-scale patterns in habitat structure that tend to occur along an elevational gradient, and the second describes finer-grained variation in woodland structure and available cover within given elevation bands. According to this interpretation, Pinyon Jays tend to occur in lower elevation bands of woodland, but within those bands they use the more open areas for caching and areas with greater cover for foraging and nesting.

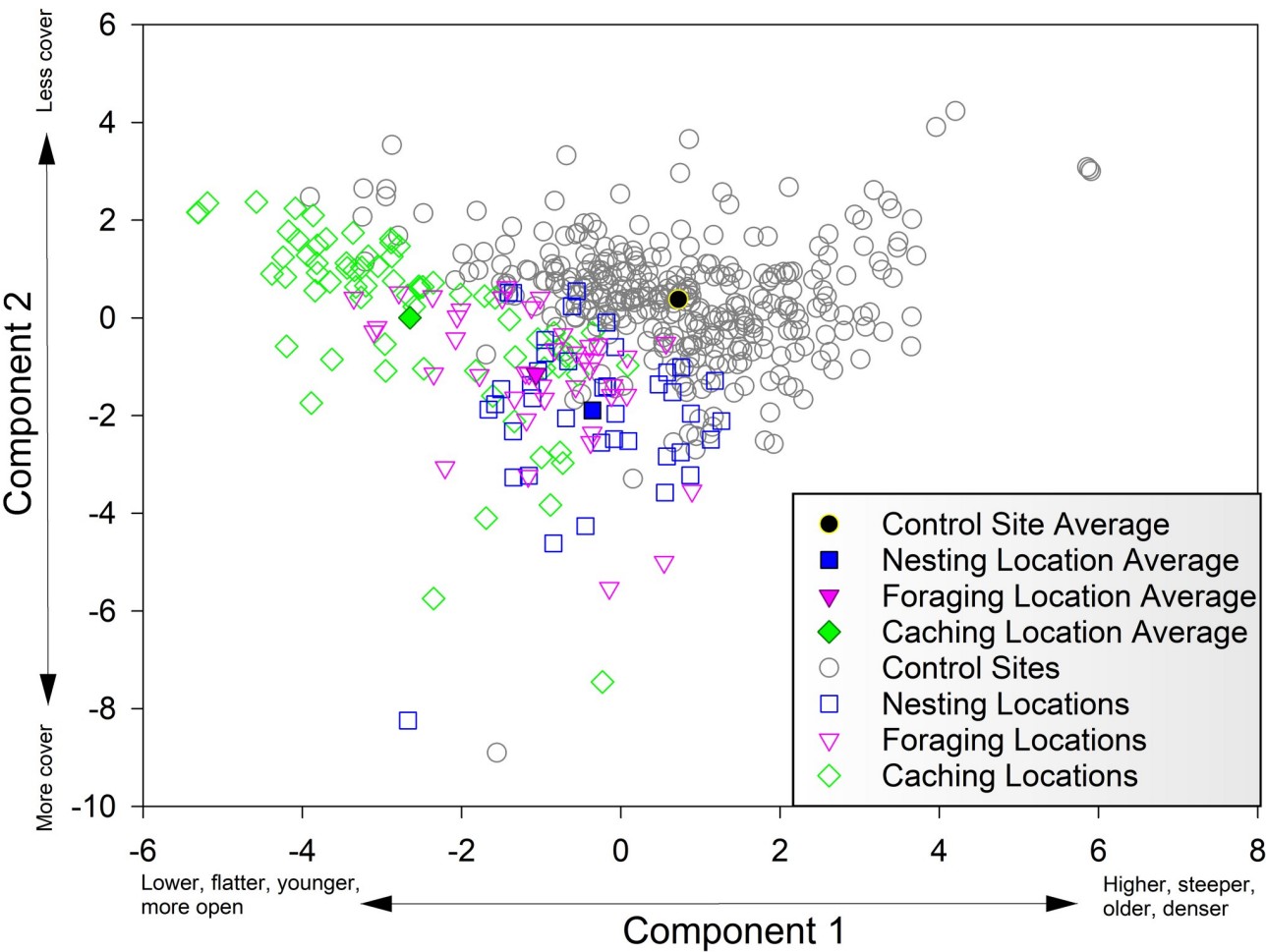

**Fig 4. Ordination plot.** Pinyon Jay locations by behavior type and control sites plotted along the first two PCA axes (see Table 4 for axis loadings). Mean values for each Pinyon Jay behavior type and for control sites are depicted as solid symbols, and individual data records are open symbols.

## Logistic regression

All three behavior-specific logistic regression models had high internal classification accuracy, averaged over all permutations; 0.890 for the caching model, 0.863 for the foraging model, and 0.920 for the nesting model. The caching model indicated that Pinyon Jay caching locations were more likely to occur with lower slope, lower tree cover, increased woody debris, shorter distance to roads, and more disturbance than control sites (Table 5). None of the predictors for Pinyon Jay foraging locations were statistically significant when averaged across all permutations due to high standard errors, but the largest effect sizes were noted for lower slope, less grass and forb cover, increased woody debris, and shorter distance to roads compared with control sites (Table 5). Pinyon Jay nesting locations were more likely to occur at lower elevations with decreased forb cover and increased woody debris compared to control sites (Table 5). Across all three Pinyon Jay behavior types, occurrence probability generally increased with lower elevation, lower slope, lower forb cover, shorter distance to roads, and increased woody debris compared to control sites (Table 5). Lower tree cover was a marginally significant predictor for caching locations but was not a significant predictor within the models for foraging and nesting locations.

**Table 4. PCA loadings.**

| Habitat Attribute (see Table 2) | Component 1 (X-Axis, Fig 4) | Component 2 (Y-Axis, Fig 4) |
|---|---|---|
| Elevation | 0.353 | -0.040 |
| Slope | 0.315 | 0.114 |
| Stand Age | 0.318 | -0.119 |
| Canopy Cover | 0.349 | -0.286 |
| Tree Cover | 0.265 | -0.298 |
| Shrub Cover | -0.200 | 0.128 |
| Forb Cover | -0.171 | 0.108 |
| Grass Cover | -0.346 | 0.186 |
| Woody Debris 1 (smallest size category) | -0.039 | -0.492 |
| Woody Debris 2 | -0.110 | -0.450 |
| Woody Debris 3 | -0.097 | -0.379 |
| Woody Debris 4 | 0.211 | 0.164 |
| Woody Debris 5 | 0.227 | 0.222 |
| Woody Debris 6 | 0.308 | 0.187 |
| Woody Debris 7 (largest size category) | 0.214 | 0.154 |
| Distance to Road | 0.187 | 0.120 |

Loadings for the first two components of the PCA.

The random effects components of the three models were all estimated to be non-zero, although the foraging model estimated a zero-value in 29% of control site allocation permutations, whereas the caching model estimated a zero-value in < 1% of permutations. The additional complexity introduced by a singularity/boundary condition (a zero-value random effect) within many foraging model permutations may have contributed to the variability (and thus reduced significance) seen in the parameter estimates.

Within the logistic regression models, we note that several predictors that had strong and statistically significant effects for a majority of permutations in the control site allocation process became non-significant when all permutations were combined. Because of this sensitivity to control site allocation, examination of data visualizations (next section) may assist in the interpretation of model results and underlying biological relationships.

## Data visualizations

Fig 5 shows box plots for all FIA attributes included in logistic regression models combined across all study regions. Fig 6 shows box plots for two additional habitat attributes (the FIA Stand Density Index attribute and the non-FIA Distance to Edge attribute) that were not included in logistic regressions or ordinations, but which could be useful indicator attributes for land managers. Notable patterns in the box plots are as follows:

1. Locations used by Pinyon Jays appeared to be a relatively distinct subset of available woodland habitat with regard to most attributes. This is consistent with patterns observed in the PCA ordination (Fig 4), and with many elements of the logistic regressions (Table 5). Additionally, locations used by Pinyon Jays for different behaviors appeared to be relatively distinct from one another, though overlapping.

2. Compared to available habitat, caching locations were concentrated in lower-elevation, lower-slope areas with younger woodland stands, lower tree cover, and generally higher but variable shrub, forb, and grass cover. The Stand Density Index and Distance to Edge

**Table 5. Logistic regression results.**

| CACHING LOCATIONS | | | |
|---|---|---|---|
| | Mean | Standard Error | p |
| Fixed Effect | | | |
| **Intercept** | **-3.687** | **1.162** | **0.001** |
| Elevation | -0.405 | 0.585 | 0.481 |
| **Slope** | **-2.066** | **0.557** | **< 0.0001** |
| Stand Age | -0.177 | 0.432 | 0.654 |
| **Tree Cover** | **-0.913** | **0.553** | **0.075** |
| Shrub Cover | 0.251 | 0.349 | 0.445 |
| Forb Cover | -0.021 | 0.3 | 0.904 |
| Grass Cover | 0.377 | 0.439 | 0.385 |
| Woody Debris | **1.235** | **0.818** | **0.010** |
| **Distance to Road** | **-2.712** | **0.789** | **< 0.0001** |
| Disturbance | **1.971** | **0.951** | **0.022** |
| Random Effect | | | |
| Eastern Nevada: Intercept | 1.838 | 0.413 | 0.0001 |
| Southern Idaho: Intercept | -1.320 | 0.301 | 0.0001 |
| Central Nevada: Intercept | -0.522 | 0.288 | 0.0001 |
| FORAGING LOCATIONS | | | |
| Fixed Effect | | | |
| Intercept | -5.486 | 7.418 | 0.044 |
| Elevation | -1.566 | 2.790 | 0.317 |
| Slope | -4.184 | 6.299 | 0.058 |
| Stand Age | -2.013 | 3.641 | 0.236 |
| Tree Cover | 1.931 | 3.079 | 0.200 |
| Shrub Cover | 1.380 | 2.479 | 0.311 |
| Forb Cover | -9.236 | 14.819 | 0.080 |
| Grass Cover | -4.970 | 8.384 | 0.113 |
| Wood Debris | 4.419 | 9.650 | 0.053 |
| Distance to Road | -3.326 | 4.848 | 0.063 |
| Disturbance | -1.700 | 75.272 | 0.555 |
| Random Effect | | | |
| Eastern Nevada: Intercept | 0.686 | 1.076 | 0.290 |
| Southern Idaho: Intercept | N/A | N/A | N/A |
| Central Nevada: Intercept | -0.707 | 1.327 | 0.290 |
| NESTING LOCATIONS | | | |
| Fixed Effect | | | |
| **Intercept** | **-4.723** | **2.623** | **0.021** |
| **Elevation** | **-2.445** | **1.600** | **0.004** |
| Slope | -1.249 | 1.090 | 0.154 |
| Stand Age | 0.528 | 1.016 | 0.564 |
| Tree Cover | 0.500 | 1.030 | 0.570 |
| Shrub Cover | 0.261 | 0.848 | 0.743 |
| Forb Cover | **-7.110** | **4.247** | **0.022** |
| Grass Cover | 0.123 | 1.256 | 0.983 |
| **Woody Debris** | **2.156** | **1.266** | **0.003** |
| Distance to Road | -1.118 | 1.291 | 0.289 |

(*Continued*)

| | | | |
|---|---|---|---|
| Disturbance | 2.606 | 2.406 | 0.115 |
| Random Effect | | | |
| Eastern Nevada: Intercept | 2.116 | 0.903 | 0 |
| Southern Idaho: Intercept | -2.088 | 0.904 | 0 |
| Central Nevada: Intercept | NA | NA | NA |

Logistic regression results over all control site allocation permutations for the caching, foraging, and nesting location analyses. Fixed effects are reported as mean and standard error of the estimate and the two-tailed p value; random effects are reported as mean and standard error of the estimate and the probability (fraction of permutations) that the term was equal to 0. Bolded terms are significant at $\alpha = 0.05$.

metrics were typically very low for caching locations. In fact, many caching locations occurred in the pure shrubland habitat located down-slope from the woodland-shrubland ecotone (note that these pure shrublands were not represented in the sample of FIA control sites).

3. Foraging locations were also concentrated in lower-elevation and lower-slope areas than control sites, but stand age, tree cover, forb cover, and grass cover were comparable to control sites. Shrub cover in foraging locations was highly variable, but tended to be higher than in control sites, as was woody debris. Compared to caching locations, foraging locations were somewhat steeper, with older stands, more tree cover, and less grass cover. With regard to the stand density index and distance from edge, foraging locations were intermediate between caching locations and control sites.

4. Nesting locations also tended to occur in lower-elevation and lower-slope areas than control sites, but to a lesser degree than foraging and caching locations. Stand age and non-tree cover were roughly comparable to control sites, but tree cover and woody debris was higher than typical control sites. As with the other Pinyon Jay behavior types, Stand Density Index and Distance to Edge were lower than control sites, but higher than caching and foraging locations. In most respects, nesting locations were intermediate between the other Pinyon Jay behavior type locations and control sites.

5. There was a distinct pattern of increasing elevation and slope, increasing stand age, increasing distance from edge, and increasing stand density moving from caching locations to foraging locations to nesting locations to control sites. Shrub cover and grass cover showed the opposite trend. Tree cover showed an increasing trend across the behavior type series, but much of this variation (with the partial exception of caching locations, many of which were in pure shrubland) occurred within the diversity of tree cover levels present within control sites. Woody debris also showed an increasing trend across the behavior type series, but most of this variation occurred within the upper bounds of what was represented within control sites.

## Discussion

### Main findings and significance

This study offers the first systematic description of Pinyon Jay occurrence patterns and behavior-specific habitat characteristics in the Great Basin. The combination of ordinations, logistic regressions, and data visualizations presented in this study collectively suggest that Pinyon Jays

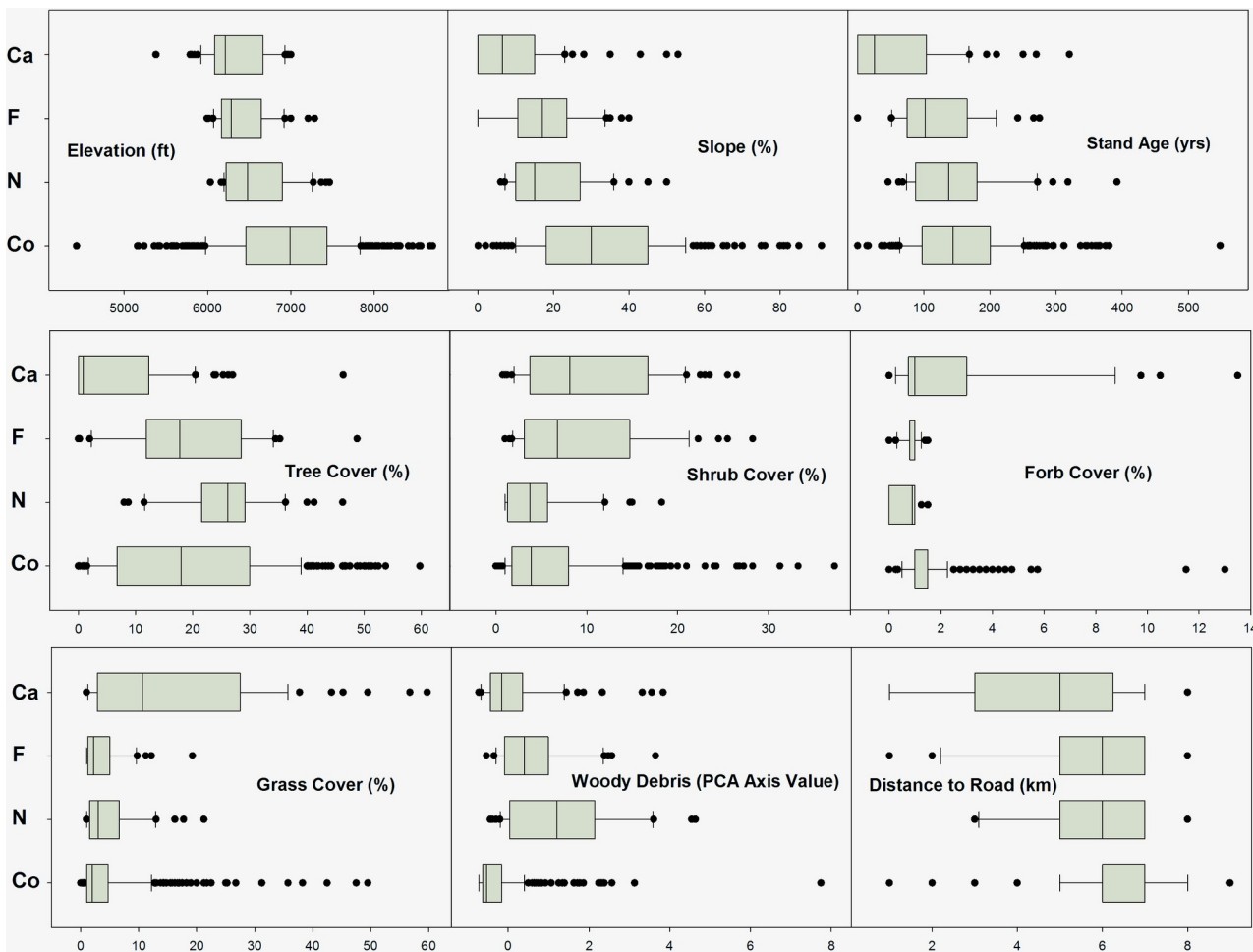

**Fig 5. Box plots for analyzed habitat attributes.** Box plots for continuous attributes used in logistic regression models, as described in Table 2. The median and lower quartile are visually indistinguishable in the Distance to Road attribute for control sites. Y-axis codes are Co = control sites, N = Pinyon Jay nesting locations, F = Pinyon Jay foraging locations, and Ca = Pinyon Jay caching locations. For better visual clarity, the extreme high range of observed Forb Cover values (15–25%) is truncated, omitting a small number of outlier Co sites and Ca locations.

use pinyon-juniper woodlands selectively and somewhat predictably. Pinyon Jay locations in our study areas were concentrated in or near lower-elevation, flatter woodlands and were less common in higher-elevation, steeper woodlands. Additionally, the areas used for different Pinyon Jay behaviors appear to have distinctive (but overlapping) habitat profiles, with caching, foraging, and nesting arrayed sequentially along gradients of increasing slope, elevation, and stand density. Similar patterns of Pinyon Jay caching and nesting activities partitioned along elevation and stocking gradients were observed by Johnson et al. [23] within a pinyon-juniper (*P. edulis* / *Juniperus spp*. association) woodland system in New Mexico, suggesting that habitat partitioning by behavior along an elevational gradient could be present across a broader physiographic range than our study regions.

Pinyon Jay caching locations were concentrated in open woodland stands with high shrub and grass cover, which are similar to the Phase I (early successional) pinyon-juniper woodlands defined in the classification scheme by Miller et al. [19], and sometimes occurred in pure shrublands. This occurrence pattern could have a mutualistic explanation [3]. From the pinyon pine perspective, seedlings likely experience less competition from established trees in

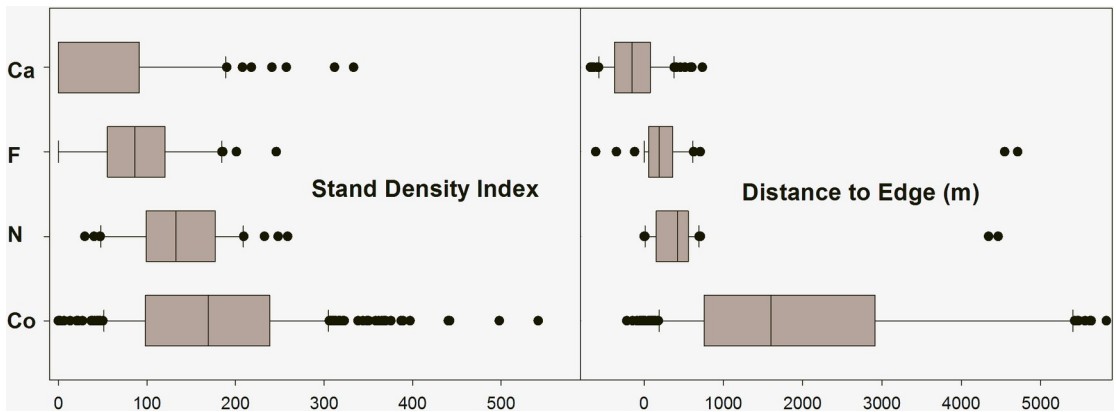

**Fig 6. Box plots for additional habitat attributes.** Box plots for two habitat attributes not used in logistic regression models, as described in Table 2. The median and lower quartile are visually indistinguishable in the Stand Density Index attribute for Pinyon Jay caching locations. Y-axis codes are Co = control sites, N = Pinyon Jay nesting locations, F = Pinyon Jay foraging locations, and Ca = Pinyon Jay caching locations. For better visual clarity, the extreme high range of observed Distance to Edge values (6,000–12,000 m) values are truncated, omitting a small number of outlier Co sites.

more open areas, and seeds placed next to shrubs, rocks, or woody debris in otherwise open areas may benefit from favorable microsite conditions created and maintained by those features [67]. From the Pinyon Jay perspective, pinyon pine seeds cached away from their source of origin may be less likely to be discovered and eaten by small mammals that specialize on pinyon seeds [68, 69].

Pinyon Jay foraging locations generally occurred in older (though still relatively young) stands than caching locations, with generally abundant woody debris across a wide range of tree cover values. Given that foraging behavior as defined in our study encompassed the gathering of diverse food items from trees, deadfall, shrubs, and ground, areas with these characteristics may offer a beneficial combination of pinyon pines in their most productive seed-bearing years [70, 71] interspersed with areas where abundant insect prey is available due to higher shrub or ground cover or woody debris [72]. The habitat characteristics of Pinyon Jay foraging locations correspond to a mosaic of Phase I and Phase II pinyon-juniper successional stages [19].

Pinyon Jay nesting locations tended to be concentrated in areas with higher tree cover and more woody debris, presumably because of the concealment they offer [23, 73]. However, like caching and foraging locations, nesting locations were concentrated in lower elevation, lower slope areas, and steeper, higher sites that otherwise offered good concealment for nesting locations appeared to be avoided. Pinyon Jay nesting locations correspond best with the denser portion of the Phase II class of pinyon-juniper woodlands, but may also include some Phase III areas [19].

All Pinyon Jay locations, regardless of behavior type, were concentrated in lower-elevation woodlands (most likely a mix of Phase I and Phase II classes) near the woodland-shrubland ecotone. Plausible explanations for this pattern could involve the longer snow-free season of lower elevations, or the presence of a mosaic of desirable habitat characteristics needed to support different behavior types. Phase I and Phase II woodlands are relatively common at lower elevations where Pinyon Jay locations are concentrated, whereas the proportion of Phase III woodlands tends to increase with increasing elevations based on our field observations. This elevational distribution of woodland Phases could be a historically-recent phenomenon [19, 22, 31–33], raising the possibility that the lower-elevation concentration of Pinyon Jay locations that we observed is correspondingly recent.

Within the lower-elevation areas where Pinyon Jay locations in our study were concentrated, jays appear to respond to a spectrum of local cover values (as indicated by the tree cover and woody debris attributes), using more open portions for caching, denser portions for foraging, and high-cover portions for nesting. This is notable because, on average, high-cover areas should be less common in the generally younger, more open woodlands of lower-elevations than elsewhere (see X-axis of Fig 4, Table 4). Collectively, this data suggests that Pinyon Jays may be selecting habitat at two different scales; a broader scale, which defines larger, home-range areas where all of their requirements are present, and a finer-scale which determines the parts of those home range that are used for specific behavior types. We note, however, that our ability to clearly demonstrate multi-scale occupancy patterns in this study is constrained by the fact that all habitat assessment data have a relatively uniform spatial resolution dictated by FIA design and protocols.

In addition to providing important information about Pinyon Jay occurrence patterns and habitat use, these findings are potentially significant for vegetation management planning and implementation because Pinyon Jays in our three study areas appear to prefer the same lower-elevation, relatively-open (at the home-range scale) woodlands where most woodland removal management is performed [74, 75]. These vegetation management projects, which are most often conducted to create or improve habitat for Greater Sage-Grouse [76] (but see Miller et al. [22] and Somershoe et al. [15] for other reasons for woodland treatments), have resulted in the removal of an estimated 45,000 ha of pinyon-juniper woodland in the Great Basin portions of Nevada, Utah, and Idaho over the last eight years alone (Witt, unpublished USFS data). Historical Pinyon Jay declines cannot reasonably be attributed to these treatments, but given the current pace and anticipated continuation of these vegetation treatments, they could be or become one factor affecting Pinyon Jay populations, either negatively or positively. Information from other regions does suggest that woodland treatments can have unintended effects on Pinyon Jays. For example, Johnson et al. [77] found that a fuels treatment within pinyon-juniper woodlands in northern New Mexico that reduced tree density by almost 90 percent prompted the local Pinyon Jay flock to avoid the treated area altogether. To date, however, almost no direct monitoring has been conducted in the Great Basin to determine if and how Pinyon Jay flocks respond to vegetation management projects that occur within or close to their home ranges.

## Interpretational considerations

The findings presented in this study should be interpreted with the following considerations in mind:

1. Pinyon Jay data were collected at the three distinct study areas that collectively encompassed 175,630 ha. Although this was a substantial area, it represents only a small portion of the Great Basin region that we would like to characterize.

2. This study combined data from three distinct projects that covered different years and seasons. This is most immediately relevant to interpreting foraging locations, given that seasonal variation in foraging behaviors and habitats are plausible. It is less likely that unstandardized seasonality of our component studies affected the interpretation of nesting and caching locations, given that these behaviors have intrinsic seasonal bounds, as previously described. Further, the FIA assessment data used in this study were collected during seasonally constrained periods.

3. We equated "potential habitat" for Pinyon Jays to all pinyon-juniper woodlands lying within 200 km of any Pinyon Jay study area. Although inferences from this study can

cautiously be extended to the pinyon-juniper woodlands beyond our immediate Pinyon Jay study areas (subject to confirmation in future studies; see Johnson and Sadoti [73] for caveats about model transferability to similar systems) it does not extend to other forest types or regions where Pinyon Jays occur.

4. Our analysis does not distinguish between flocks within a study area, and does not allow us to draw inferences about inter-flock variability with regard to behavior-specific occurrence patterns.

5. The iterative allocation of control sites among Pinyon Jay behavior types in logistic regression modeling was necessary, but it could have diluted the statistical significance of some important predictors of occurrence. Future modeling efforts where the spatial extents and sample sizes of Pinyon Jay data and control data are better matched should reduce this issue. Models created from data sets with a large sample of Pinyon Jay locations will also allow data to be withheld from the model building process and used for external validation.

6. The relationship between Pinyon Jay occurrence and distance to road is potentially non-causal. Road density is typically higher in lower-elevation, flatter areas (i.e. valley margins) than at higher-elevations, and Pinyon Jays could, and probably do, prefer these lower areas for reasons completely unrelated to the proximity of roads. However, one author has suggested the vegetation typically present alongside graded, unpaved roads may provide valuable foraging opportunities for Pinyon Jays [78].

7. The Distance to Edge attribute used for data visualizations shows clear contrasts across behavior-specific Pinyon Jay locations and control sites, but it needs to be further investigated using more formal methods for delineating ecotones. Additionally, the patterns seen in our data (Fig 6) were emphasized because control sites by definition excluded the pure or near-pure shrublands that comprised a significant proportion of Pinyon Jay caching locations, and a smaller proportion of foraging locations.

Currently, we are analyzing a separate Pinyon Jay data set derived from a long-term statewide bird monitoring program in Nevada. Because these data were obtained from a broader-scale fully randomized sampling design and used a standardized survey protocol, they should provide a useful independent characterization of Pinyon Jay occupancy patterns in the Great Basin.

## Conclusions and recommendations

Pinyon Jay populations have been declining precipitously for at least the last half-century, while the pinyon-juniper woodlands that they inhabit in the Great Basin are thought by many to have been expanding at unprecedented rates [19, 22, 31–33, 79–83]. Given this apparent paradox, identifying the reasons for Pinyon Jay declines is critical for defining constructive conservation actions that ensure the species' long-term viability [3, 15]. This urgency is particularly important given the widespread and potentially accelerating woodland management activities in the Great Basin that prioritize creation or preservation of shrublands without a definitive understanding of their effects on Pinyon Jays. Progress towards a more inclusive management paradigm can be achieved through: a) better knowledge of Pinyon Jay ecology and habitat requirements, b) monitoring of management impacts, c) better understanding of the ecology and dynamics of the woodlands that comprise Pinyon Jay habitat, and d) integration of knowledge obtained from these three areas into existing vegetation management protocols and guidance.

A fundamental need is for more robust, spatially-extensive data characterizing Pinyon Jay occurrence patterns and habitat use as a function of region, season, and behavior type, both

across and within the individual flock level. Ideally, future studies can incorporate habitat descriptors at multiple scales, given the possibility that Pinyon Jay habitat selection operates at more than one scale (see above). To maximize their integrative value, these data sets would ideally be gathered using a standardized survey protocol. Pinyon Jays, however, present multiple challenges to the field biologist, study designer, and data analyst [15], and approaches suitable for typical passerine birds may be suboptimal for Pinyon Jays for several reasons. Unlike species where a single breeding pair occupies a clearly defined territory at all times during the breeding season, Pinyon Jays occupy (and presumably select) habitat at the flock and subflock level. Pinyon Jays are also year-round residents, and protecting breeding habitat alone may be insufficient for effective conservation. The Pinyon Jay's pattern of habitat use, which involves flock movements across a relatively large home range to accommodate different behaviors and take advantage of seasonally-varying food resources, has potentially important effects on both the detection properties of a given survey protocol and the ecologically-legitimate interpretation of resulting data. As a simple example, a Pinyon Jay flock may be frequently absent from a critically important subset of its home range, either during portions of the day, or during entire seasons. Similarly, roaming flocks may frequently fly over or loaf in areas of the home range that are not critical to home range quality or viability. To provide accurate and actionable information about the habitat requirements of Pinyon Jays, survey protocols and research study designs need to account for these realities appropriately, operate at scales that reflect actual Pinyon Jay habitat selection patterns, and be guided by a sampling framework that produces well-balanced data suitable for presence / absence modeling. The multi-agency Pinyon Jay Working Group [15] is currently exploring options for standardized Pinyon Jay survey protocols. The same group's Pinyon Jay Conservation Strategy [15] also notes that some of the information needed to better characterize Pinyon Jay habitat requirements could be obtained by systematically monitoring Pinyon Jay responses to vegetation management activities, especially in situations where their pre-treatment presence has been confirmed by baseline or clearance surveys.

With regard to pinyon-juniper woodlands, their structural attributes and other characteristics that might be limiting to Pinyon Jay populations need to be further studied. We suggest that it may be especially important to identify the correlates or profiles of tree stands and landscapes that exhibit a predictable and/or abundant pinyon pine mast [22]. Our preliminary review of FIA data collected in Nevada between 2006–2015 suggests that woodlands matching the structural characteristics Pinyon Jays used for foraging in this study are 5–7 times less extensive than nesting or caching habitat in Nevada (unpublished USFS data). Presence of reliably productive stands within the home range could be especially important to Pinyon Jays during years of more generally depressed pine mast production. Given evidence of reduced mast production in pinyon pine [83], and associated changes in habitat use by Pinyon Jays [24] in some areas affected by climate change, it might be critical to long-term Pinyon Jay conservation to systematically investigate the quality and quantity of good foraging areas.

In addition, continued research and communication is needed to better clarify the degree to which woodland expansion and infill is part of a historically "normal" dynamic, versus a problematic departure condition, and how to distinguish between these phenomena on the landscape. Without this more holistic understanding, colonization of shrublands and other open areas by trees tends to be regarded as "invasive" by default, even though at least some of these more-recently colonized areas may provide important Pinyon Jay habitat in the Great Basin. Achieving this broader perspective may be a necessary prerequisite to successfully accommodating the needs of Greater Sage-Grouse, Pinyon Jays, and other sensitive shrubland and woodland bird species within the overall framework of pinyon-juniper woodland management.

Ultimately, accruing information about Pinyon Jays should be incorporated into woodland management paradigms and protocols in the Great Basin (see Ricca et al. [84] for an example) in ways that accommodate both previously-identified and newly-emerging goals within the context of healthy ecosystem function [85] and landscape diversity. For the present, Somershoe et al. [15] provide guidance for managers seeking to incorporate Pinyon Jay conservation measures into their vegetation management projects based the extent of current knowledge.

## Supporting information

**S1 Table. Full data set.** All data used for data visualizations, ordinations, and logistic regression analysis. Each record refers to either a behavior-specific Pinyon Jay location or a control site. Each record includes various locational and data category attributes, including decimal latitude and longitude, along with all habitat attributes considered for inclusion in analysis, as described in Table 2. All attribute headers are sufficiently explicit to be self-explanatory when viewed in conjunction with Table 2. Definitions of codes for habitat types (which were not used in any analysis or data visualization) are available in Alexander 1988 [60]. Missing data are intrinsic to the FIA data set.
(XLSX)

## Acknowledgments

Field work was conducted by Gustavo Gonzalez, Michael Maples, and John B. Free (in Eastern Nevada), by Murrelet Halterman, Natasha Peters, Larry Teske, and Mercer Owen (in Southern Idaho), and by Mercer Owen and Sue Brunner (in Central Nevada). Their data collection efforts, which were supplemented by the authors, are much appreciated. Statistical analysis was performed by Juniper Simonas of Dapper Stats (www.dapperstats.com).

Wallace Keck, Superintendent of City of Rocks National Reserve and Park Manager for Castle Rock State Park, provided us with extensive information about local Pinyon Jay populations. LeAnn and Kim Draper allowed their property to be used as a capture location and staging area for telemetry efforts. At the Bureau of Land Management, Sandra Brewer and John Wilson provided invaluable assistance and support in Central Nevada. We thank all of these individuals.

## Author Contributions

**Conceptualization:** John D. Boone, Elisabeth M. Ammon.

**Data curation:** John D. Boone, Chris Witt.

**Formal analysis:** John D. Boone, Chris Witt.

**Funding acquisition:** John D. Boone, Chris Witt, Elisabeth M. Ammon.

**Investigation:** John D. Boone, Chris Witt, Elisabeth M. Ammon.

**Methodology:** John D. Boone, Chris Witt, Elisabeth M. Ammon.

**Project administration:** John D. Boone, Elisabeth M. Ammon.

**Resources:** John D. Boone, Chris Witt, Elisabeth M. Ammon.

**Software:** John D. Boone.

**Supervision:** John D. Boone, Elisabeth M. Ammon.

**Validation:** John D. Boone.

**Visualization:** John D. Boone.

**Writing – original draft:** John D. Boone, Chris Witt.

**Writing – review & editing:** John D. Boone, Chris Witt, Elisabeth M. Ammon.

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
