## [Decision Letter · Decision Letter 0]

9 Oct 2020

PONE-D-20-21672

Behavior-specific occupancy patterns of Pinyon Jays (Gymnorhinus cyanocephalus) in three Great Basin study areas and significance for pinyon-juniper woodland management

PLOS ONE

Dear Dr. Boone,

Thank you for submitting your manuscript to PLOS ONE. After careful consideration, we feel that it has merit but does not fully meet PLOS ONE’s publication criteria as it currently stands. Therefore, we invite you to submit a revised version of the manuscript that addresses the points raised during the review process.

We look forward to receiving your revised manuscript.

Kind regards,

Bi-Song Yue, Ph.D

Academic Editor

PLOS ONE

2. In your Methods section, please provide additional location information of the study area, including geographic coordinates for the data set if available.

3. We note that Figures 1, 2 and 3 in your submission contain map/satellite images which may be copyrighted.

a. You may seek permission from the original copyright holder of Figures 1, 2 and 3 to publish the content specifically under the CC BY 4.0 license. 

Reviewers' comments:

Reviewer's Responses to Questions

**Comments to the Author**

1. Is the manuscript technically sound, and do the data support the conclusions?

Reviewer #1: Partly

Reviewer #2: Yes

2. Has the statistical analysis been performed appropriately and rigorously? 

Reviewer #1: No

Reviewer #2: Yes

3. Have the authors made all data underlying the findings in their manuscript fully available?

Reviewer #1: Yes

Reviewer #2: Yes

4. Is the manuscript presented in an intelligible fashion and written in standard English?

Reviewer #1: Yes

Reviewer #2: Yes

5. Review Comments to the Author

Reviewer #1: This study provides some interesting behavioral ecological data on habitat choice by behavior in pinyon jays inhabiting pinyon – juniper forest and scrubland. The study encompasses three representative but widely separate locations, and creatively uses US Forest Service forest inventory data for ecological characterization of sites.

While the study is creative and interesting, I found some of the statistical analysis methodologies hard to follow, and I have a number of questions or concerns that should be addressed. One frustration is that there are differences in the timing and some methodologies among the data collected at the three sites, making it hard to interpret any apparent differences in results among them.

1) The data are observational or observational with aid or radiotelemetry. Basically research techs did focal group follows during daylight hours, and periodically recorded behavior and location.

a) Behavior – location data were recorded “approximately once per hour” (line 188). Why wasn’t this standardized? There is always a risk with observational data that you record data more frequently where it’s easiest to see the birds.

b) What was done to avoid sample biases due to observability and accessibility? This can potentially be a big problem with observational data collection. Foraging and cacheing were observed more frequently in open (less dense) pinyon-juniper and flatter areas than expected based on a random selection of points within the region. But wouldn’t this be precisely the areas where the birds are easiest to detect and observe?

c) It appears that the birds are active closer to the road than expected by chance. Again, might this not be because the groups that were selected for focal observation and the easiest and quickest detection is near roads?

2) I have a lot of concerns about the logistic regressions.

a) I don’t understand this process of pairing ‘control’ FIA sites - really background reference sites - with sites used by the jays. Why not use all? It is not necessary that there be equal numbers of jay-use and reference sites. This seems unnecessarily complicated, and added a lot of instability to the modeling.

b) If I understand this correctly, the overall logistic regression model was not significant. If not, there is no reason to evaluate the significance of each predictor.

c) The usual approach on an exploratory statistical data analysis with data like these would be to compare different models (all models or subsets) using AIC. Doing so, it would be possible to evaluate which predictors or combination of predictors do better than the intercept null model. Here is a good example:

Ross, A.M., Johnson, G. and Gibbs, J.P., 2016. Spruce grouse decline in maturing lowland boreal forests of New York. Forest Ecology and Management, 359, pp.118-125.

3) Table 3: It is unclear what the numbers represent. Is Eastern Nevada # caching sites = 12 (106) mean there are 12 locations that were detected being used by the two flocks and 106 observations of use of the 12 sites? The 0(0) entries should be NA, at least in some cells correct? If the southern Idaho data are from outside the nesting season, that nesting sites is NA.

4) Line 588: There has been an apparent long-term decline in pinyon jays. The authors suggest it has to do with loss due to land management practices of pinyon-juniper stands that have the characteristics that jays prefer for foraging and caching. Is 45,000 ha removal in the last 8 years a lot in relation to pinyon jay habitat? It doesn’t seem so, given the large distribution in the American Southwest. This would equal habitat for how many jays / jay flocks? Is there other evidence of enough habitat loss to account for pinyon jay population declines?

5) Lines 633-635: What does this mean? Why were these excluded? Is it because they don’t occur in the FIA data? I didn’t see this explained in the Methods, and would seem to have a serious impact on the data interpretation.

6) The ordination is interesting and appears appropriately done.

Reviewer #2: Overview

This is a timely and potentially impactful study. It is well described throughout. There are some limitations to the study design beyond the authors control, but nevertheless it a novel and useful paper. I have some queries/comments about the analytical methods, in particular the logistic regression.

Detailed comments

Figure 1-3: I would suggest you add the state names to these figures and consider whether to have an overview inset of the states in figure 1 with a rectangle showing the area in the map.

Fig4: not sure this is required, the survey design is well explained in the text.

121:124: ‘Control sites were not assumed to be unoccupied by Pinyon Jays, but were intended to provide a representative characterization of the diversity of pinyon-juniper woodland habitat available within the general study region.’

This statement makes your study sound more like a use-availability study rather than a case-control? In believe in case-control studies the control sites are known to be unoccupied.

Table 2: I understand that for logistical reasons the different regions were surveyed at different times of year as were the control sites. However, it is worth some consideration of how this might influence your results. Many of the co-variates in the table will remain constant over the year, but some, e.g. % forb and %grass cover, I imagine would vary?

342: Why did you chose to do the ordination separately for each region? It would be useful here to see if the different behaviour types, or control sites, cluster together regardless of region. This could inform the logistic regression.

361: ‘Because the ratio of Pinyon Jay locations to control sites was low, the odds ratio output from logistic regression can be treated as an approximation to the resource selection function [42, 67, 68].’

It is my understanding that under certain circumstances use-availability models can approximate the resource selection function. I thought this was when the probability of use was low across all the range of co-variates, rather than when the ratio of use to available sites in the model was low?

374:376: You may be right but it would be good to know why you don’t want to ‘re-use’ control sites in the different behaviour models? This seems to limit the number of control sites you can have, which several authors suggest should be maximised.

Also, if I understand this correctly you end up with quite different ratios of use to control sites in the different models, specifically with very few control sites in the Foraging model. Also, all models will be dominated by control sites from Eastern Nevada, as there are more of them.

An alternative approach would be to define the ratio of use to control sites you want and then select the appropriate number of control sites at random from the regions in that model.

379: Logistic regression, would be good to spell out the structure of model. This is apparent from Table 4, but it would be good to clarify in the methods.

379: I am not convinced about having ‘region’ as a random effect as it only has two or three levels depending on the model. It is usually recommended that a random effect have at least six levels and ideally considerably more as it becomes difficult to estimate the model parameters below this. I suspect that is one reason why some of your models didn’t converge and why you got a lot of zero estimates for the random term in the foraging model (469:474). I would suggest you treat ‘region’ as a fixed effect or if you do pool the regions in the ordination and the control sites show considerable clustering you could consider dropping region from the model, which would give a far bigger pool of control sites to select from in each iteration.

379: For nesting model, would it be better to select control sites surveyed at the relevant time of year? I ask this as I see that % forb has some effect and this is likely to vary seasonally. If this is not possible due to the available sample of control sites, it is worth some discussion.

6. PLOS authors have the option to publish the peer review history of their article (what does this mean?). If published, this will include your full peer review and any attached files.

Reviewer #1: No

Reviewer #2: **Yes: **F Burns

---

## [Author Response · Author response to Decision Letter 0]

15 Dec 2020

RESPONSES TO REVIEW COMMENTS 

Thank you to both reviewers for your very thoughtful comments. We have taken them very seriously, believe that they have improved the manuscript substantially. We do our best below to respond to each reviewer comment in detail with the same degree of thoughtfulness. All original review comments are reproduced in below, followed by our responses. 

Reviewer #1

One frustration is that there are differences in the timing and some methodologies among the data collected at the three sites, making it hard to interpret any apparent differences in results among them.

We fully agree that it would have been best to have all critical parameters entirely regularized over the three studies, particularly seasonality, along with a full array of behavioral types for each study area. However as described in the manuscript, seasonality varied among the three study areas because of the requirements and timing of different agency funding agreements (which were beyond our control), which in turn imposed one of the two behavior-type “data gaps” (see Table 3) given that nesting locations could not be recorded in the study area that was surveyed outside the nesting season. Despite this shortcoming, we believe the three data sets are considerably stronger presented together than separately, given that they do show several consistent and potentially-important patterns despite the potentially confounding influence of season. We note further that the confounding influence of season is probably less problematic than it might initially appear, for these reasons: 

1) The nesting behavior type, which is captured in two of the three study areas, is intrinsically constrained by season. In other words, we are not measuring nesting in different seasons were it might manifest differently. Rather, in the places where nesting was recorded, it was seasonally-standardized by default. 

2) The caching behavior type was detected in all three study areas, one of which was surveyed in a substantially different time of year than the other two study areas. In principle this could confound resulting data. However, it is important to emphasize (as is noted in Table 1), that the caching behavior type involves both the placement of food items, and the retrieval of those food items at some later date. Decisions about where to cache nuts are made primarily in the fall, when pine nuts are harvested. Retrieval of those nuts may occur in other seasons, but in a spatial sense, they still reflect the same, original, seasonally-constrained habitat choices that were made when the nuts were first cached. Thus even where this behavior type is recorded in different seasons, it is plausible to suspect that it largely reflects habitat choices that are more seasonally-standardized. The fact that patterns associated with caching behavior were fairly consistent across all three study areas tends to support this idea. 

3) Habitat assessments of Pinyon Jay locations (all behaviors) and control sites using the FIA protocol WERE seasonally regularized in all cases, and took place within the seasonal time window allowed by the FIA protocol, regardless of the season in which Pinyon Jay data were collected. 

Therefore, the only significant element of data collection that was not at least to some extent seasonally controlled was the foraging behavior type, which was recorded at two study areas in two different seasons. It is entirely plausible (as originally mentioned in the manuscript, with that description now strengthened in this revision) that foraging habitats do shift seasonally, and sure enough, the results we obtained for the foraging sites appear to be less well-defined and generally “fuzzier” (in both ordination and logistic regression) than for the other two major behavior types we addressed, although they were still relatively distinct from the mass of control sites. We considered breaking down the distribution of foraging locations by season analytically, but that is problematic in that each season is then represented by only a single study area. Therefore, we elected to leave the current description of foraging habitat “as is” (in its relatively fuzzy form), rather than risking reporting a spurious distinction between fall and spring foraging based on the limited information at hand. 

Overall, despite the limitations noted by the reviewer stemming from inherent imperfections in the study design, we believe that this work does illustrate some relatively strong and robust patterns of interest, and that the positives of combining the three study areas in analysis outweighed the negatives. 

Location data were recorded “approximately once per hour” (line 188). Why wasn’t this standardized? There is always a risk with observational data that you record data more frequently where it’s easiest to see the birds.

This was standardized to the extent practicable. Our goal and our instructions to surveyors were to attempt to obtain locations once per hour. However, because following a flock of moving Pinyon Jays over an often rugged landscape requires surveyors to hike and then locate workable vantage points on an ongoing basis, there is no guarantee that they will be in a position to make observations at exact one hour intervals. Even where telemetry was used, the process of acquiring new fixes could require an unknown amount of time and effort for each data collection point, and getting into a position to observe behaviors by eye could require more time. Thus, we did specify an “ideal” standardization, but recognized that the reality of the field situation would make it impossible to adhere to that standard exactly. Because of this, acquiring a data point that did not adhere exactly to one-hour intervals was not discouraged if it was possible, given that there was no way to know whether the flock would be observable at the upcoming exact hourly interval. Our main intent in specifying the approximate timing of data collection was to ensure that activities over the whole diurnal period were represented more or less evenly in the data sets, and we also had additional field and post-processing safeguards in place to ensure that the data sets used for analyses were not seriously biased towards the areas where the flocks were easiest to observe. First, in the field, as noted in the manuscript, as the data gathering surveys for a given flock proceeded over days and weeks, effort to obtain locations was increasingly directed towards to any portion(s) of the diurnal cycle that were under-represented in the data collection to date. Because Pinyon Jay flock activities were very predictable on a day-to-day basis within a given season, this had the effect of ensuring that extra effort was devoted to obtaining locations in any parts of the home range that were not easily observable and therefore under-surveyed during the early stages. Second, in the data post-processing phase, the data thinning process described in the manuscript prevented us from committing “within-flock-pseudoreplication”. Pre-thinned (raw) data did have instances where multiple data points represented the same flock, doing the same thing, in approximately the same place, and had all these data points been used in analyses, it could have indeed over-leveraged the importance of the more easily-observed locations. However, the post-processing weeded out these spatio-behavioral replicates fairly aggressively. Finally, we noted that the data sets actually used for analysis after post-processing represented the diurnal cycle fairly evenly, suggesting the potential for over-interpreting subsets of the home range was limited. 

What was done to avoid sample biases due to observability and accessibility? This can potentially be a big problem with observational data collection. Foraging and caching were observed more frequently in open (less dense) pinyon-juniper and flatter areas than expected based on a random selection of points within the region. But wouldn’t this be precisely the areas where the birds are easiest to detect and observe?

This is absolutely a valid concern, and one that we raised explicitly at some length in the manuscript (both Methods section and Discussion section). We thought quite a lot about this issue over the course of this field work once we first began to notice the apparent preference of Pinyon Jays for more open, lower-elevation areas. In short, we cannot guarantee the absence of any observational bias, but we did take critical and systematic steps to minimize its potential impact. These are as follows. 

1) At the among-flock level, as described in the paper, we did initial searches flexibly but systematically to ensure that all, or nearly all flocks present in a designated area (each of which encompassed a broad elevational gradient) were detected. Pinyon Jays flocks are easy to see and easy to hear at significant distance, so if skilled field technicians are systematically searching a designated area that is used by Pinyon Jays, we believe that the chance that they will completely fail to detect a flock over an extended preliminary survey period is very low. This initial survey step helped to ensure that were not merely selecting the easiest flocks to find for detailed study, but rather selecting all flocks that were persistently present in a given area.

2) At the within-flock level:

a. As described in detail in our response to the review comment immediately above, by ensuring that observations recorded represented the entire diurnal period of each flock studied, and that repeated observations of a flock in the same place doing the same thing only “counted once”, we greatly reduced the opportunity to inappropriately emphasize only the “easy” observations. This premise is based on the observation that Pinyon Jay flock activities were highly predictable on a day-to-day basis, which in our work has appeared to be almost invariably the case. 

b. Similarly, increasing the field effort’s focus on the times of day with fewer data points as data collection proceeded for a given flock (see response to previous item) tended to create a final data set that was reasonably balanced by time of day. 

c. The flocks followed by radio telemetry (Table 3) (which was in theory less prone to observational bias) did not appear to have appreciably different pattern of habitat use than the flocks followed only with observation. 

d. During observational surveys, initial contact with flocks was often obtained aurally at distances of up to 500 m or more. Aural detections are likely to be far less prone to a “vegetation density” bias than visual detections. 

We have taken a few opportunities in the revised text to more clearly emphasize the valid concern noted by the reviewer and the steps we took to limit its potential effects. However, at the end of it all, observational studies of this type are not able to completely negate to possibility of some observational bias. We have noted in the Discussion specifically that this possibility exists and that it needs to be evaluated critically in future studies. 

As a possible point of interest to the reviewer, we are currently analyzing data on Pinyon Jay occurrence over the 20-year period covered by our Nevada Bird Count program, which consists of point counts that randomly sample the landscape without the biases of an opportunistic observational study. So far we are seeing the same general pattern whereby Pinyon Jays are more likely to be detected in the lower elevations of PJ woodland, near where the shrubland ecotone exists. Whether this pattern holds outside the Great Basin remains to be seen. 

It appears that the birds are active closer to the road than expected by chance. Again, might this not be because the groups that were selected for focal observation and the easiest and quickest detection is near roads?

See above for general response to the issue of observational bias. In our study areas, and probably in most places, distance to road was very well-correlated to “flatter and lower”, since roads were present at much greater density in these areas than in higher, steeper terrain. Thus, an equally plausible explanation for an inverse relationship between Pinyon Jay occurrence likelihood and distance to road is that both Jays and roads tend to occur in these areas for unrelated reasons. Interestingly in the revised ordination (see reviewer 2 comment and reply, below), the distance to road attribute only “lighted up” when you get down to the 4th component of the PCA, whereas all the other attributes with which it was most notably correlated lit up in component 1. 

I have a lot of concerns about the logistic regressions.

On reflection, we acknowledge that we contributed to this concern by over-streamlining the description of analytical methodology that was originally provided by our statistician. That additional detail has been re-inserted in the revision. More specific responses to the elements of concern are below. 

I don’t understand this process of pairing ‘control’ FIA sites - really background reference sites - with sites used by the jays. Why not use all? It is not necessary that there be equal numbers of jay-use and reference sites. This seems unnecessarily complicated, and added a lot of instability to the modeling.

To clarify, there was no pairing (in the statistical sense of the word) of control sites with PIJA sites on a one-to-one basis, and there were no “equal number” constraints imposed. In the revision, we have sought to greatly improve our description of what WAS done (and why), which we fully concur was inadequately described and therefore confusing in the earlier version. The gist of what we did for this analysis was as follows. 

1) Regional attribution of control sites: The control sites that were available from the pre-existing FIA data set that met the stated criteria for inclusion were given a regional attribute that corresponded to the nearest of the three Pinyon Jay study areas. This was a necessary step in order to include a regional dimension in the logistic regression analysis. Biologically, it also made sense in that “available” habitat for each of the three study areas would then be characterized using control sites that were in the same general area, rather than control sites that might be located hundreds of miles away. As shown in Table 3, the number of control sites attributed to each region was not constrained to be equal, and in fact varied from 312 sites for Eastern Nevada to 53 for Central Nevada. This was because the only criteria for regional attribution of control sites was the closest Pinyon Jay study area, and because the distribution of pre-existing FIA sites with the appropriate characteristics was not uniform. 

2) Apportionment of available control sites by behavior type: Three different logistic regression models were undertaken, one for each of the main behavior types studied. Within each of these models, region was included as a random effect (see response to reviewer #2 below for discussion about this). Using all available control sites for each of these three models sequentially would have artificially inflated Type 1 error. To avoid this pitfall, we used subsets of available control sites for each of the three models so than none of the control sites were “reused” across the three models within any particular model run cycle. Further, we iterated this process many times, with the composition of subsets varying randomly with each iteration. This iteration process smoothed over any variations stemming from unique permutations of the control site allocation in any single iteration. Additionally, the iteration process also had the more subtle advantage of enabling us to “extract” information from all of the regionally-appropriate control sites (averaged over the multiple iterations) for each of the three models without risking inflated type 1 error. The size of the subset of control sites used for a given type of behavioral model was always fixed (though the array of specific control sites that comprised that subset varied with each iteration), and Table 3 has been revised to make that very much clearer. This fixed subset size was proportional to the frequency of the three PIJA behavior types within a given region, multiplied by the total number of control sites available for that same region. As an example, there were 53 control sites with the “Central NV” regional attribute. Within the Central NV PIJA study area, there were 76 PIJA use records used in the analysis after data thinning; 34% of these were caching, 29% were foraging, and 37% were nesting. Thus, in any given modeling iteration, 0.34 x 53 = 18 of the available control sites were used for the caching model, 0.29 x 53 = 15 for the foraging model, and 0.28 x 53 = 20 for the nesting model. 

If I understand this correctly, the overall logistic regression model was not significant. If not, there is no reason to evaluate the significance of each predictor.

The logistic regressions for caching and nesting are “significant” by virtue of having significant (i.e. non-zero) coefficients for one or more of the fixed effects. There isn’t a single “overall” evaluation of whether the entire model is significant. Rather, the whole model can judged in any of several ways, including the classification accuracy summary that we report in the results. In addition, it is generally considered to be biologically informative to evaluate terms, especially when they are “nearly significant”. 

The usual approach on an exploratory statistical data analysis with data like these would be to compare different models (all models or subsets) using AIC. Doing so, it would be possible to evaluate which predictors or combination of predictors do better than the intercept null model. Here is a good example:

Ross, A.M., Johnson, G. and Gibbs, J.P., 2016. Spruce grouse decline in maturing lowland boreal forests of New York. Forest Ecology and Management, 359, pp.118-125.

We agree that the approach that is suggested (model building and comparison using AIC) is a common and valid approach, and one that we have used previously. It is equally common and valid to build and evaluate a single model using the approaches we describe in the manuscript, especially when we have some biological “sense” of the system in question based on familiarity and observation. For example, this is the approach suggested by many standard texts, including Gotelli and Ellison 2004. In this case, we felt that given the substantial real-world consequences of throwing a “Pinyon Jay monkey wrench” into the existing paradigms of vegetation management in the Great Basin, evaluating many models to come up with a “good” one would be less compelling to readers than being able to illustrate that a single, reasonably designed model returned some significant results. In short, our feeling is that both approaches are fully valid and common, and we had practical reasons to favor the single model approach in this case. In future manuscripts, where Pinyon Jay occurrence is evaluated using GIS-derived predictors with the aim of creating an occurrence likelihood layer, we plan to use a model-comparison approach to optimize the resulting SDM. 

Table 3: It is unclear what the numbers represent. Is Eastern Nevada # caching sites = 12 (106) mean there are 12 locations that were detected being used by the two flocks and 106 observations of use of the 12 sites? The 0(0) entries should be NA, at least in some cells correct? If the southern Idaho data are from outside the nesting season, that nesting sites is NA.

We agree that as originally presented, this table was cluttered, confusing, and easy to misinterpret. We have edited the caption and substantially re-worked the table to make it easier to interpret. To answer the question posed, in this table as originally presented, the numbers mean that that are 12 recorded caching locations in Eastern Nevada (after eliminating any records that were duplicative per the discussion in the Methods section), and that these 12 Pinyon Jay caching sites were analytically paired with 106 “control” sites in each permutation of the logistic regression model. The specific 106 control sites used (out of the 212 control sites that had a regional attribute of “Eastern Nevada”) varied randomly with each iteration, but they always totaled 106 of these 212. We agree with the comment about replacing some zeroes with “N/A”.

Line 588: There has been an apparent long-term decline in pinyon jays. The authors suggest it has to do with loss due to land management practices of pinyon-juniper stands that have the characteristics that jays prefer for foraging and caching. Is 45,000 ha removal in the last 8 years a lot in relation to pinyon jay habitat? It doesn’t seem so, given the large distribution in the American Southwest. This would equal habitat for how many jays / jay flocks? Is there other evidence of enough habitat loss to account for pinyon jay population declines?

To clarify, we absolutely do NOT believe that PJ removal is the likely cause of decades of documented Pinyon Jay decline, for several obvious and less-obvious reasons, including the one stated by the reviewer. We do state clearly the more plausible explanations for the long-term decline (lines 80 – 93). We also mention that PJ removal is a dominant vegetation management objective in the Great Basin (lines 96 – 100) and that its effects on Pinyon Jays are unknown, but do not “blame” it for long-term declines. In the Discussion, we did note that PJ removal is occurring at a scale where it could be, or could become “a” factor contributing to Pinyon Jay declines, and we have slightly modified this passage to make clearer that we don’t attribute historical declines to this factor. That said, we do continue to emphasize that it is important to examine current vegetation management practices vis-a-vis Pinyon Jays so that those practices: a) don’t contribute to this decline, and b) can hopefully be designed to help to slow or reverse the decline.

Lines 633-635: What does this mean? Why were these excluded? Is it because they don’t occur in the FIA data? I didn’t see this explained in the Methods, and would seem to have a serious impact on the data interpretation.

That’s correct, distance to edge was not included in the logistic regression analyses or ordination because it was not part of the FIA data set, and therefore an “oddball” attribute. Upon seeing the results of ordinations, LRs, and data summaries of the analyzed attributes, we wondered if there were perhaps some good “indicator” attributes that somehow would capture the combination of elevation, flatness, and woodland openness that seemed to be important. The two we came up with were distance to edge (calculated by us), and the FIA Stand Density Index. The latter was not included in original analyses because it was a derived index created out of more fundamental FIA data that were used in analyses. These two potential indicator attributes were presented in a separate figure in the form of box plots because we felt they might help managers to get a better synthetic sense of how Pinyon Jays appear to use the landscape in our study areas, and presumably the somewhat broader region. We have made some changes in the revision to make clearer the rationale for presenting these two attributes in the way that we did, while excluding them from analyses. 

Reviewer #2

Figure 1-3: I would suggest you add the state names to these figures and consider whether to have an overview inset of the states in figure 1 with a rectangle showing the area in the map.

This has been done, with one exception. We don’t believe the inset is needed on figure 1, because the figure caption has been edited to explicitly state that this figure shows the western United States, which should provide sufficient locational context. State name abbreviations have been added in all cases. 

Fig4: not sure this is required, the survey design is well explained in the text.

We agree. Figure 4 has been omitted.

121:124: ‘Control sites were not assumed to be unoccupied by Pinyon Jays, but were intended to provide a representative characterization of the diversity of pinyon-juniper woodland habitat available within the general study region.’ This statement makes your study sound more like a use-availability study rather than a case-control? In believe in case-control studies the control sites are known to be unoccupied.

After consultation with our statistician, the original statement from the manuscript as quoted above is indeed misleading, and it has been replaced with a statement that is a more accurate representation of the assumed status of control sites. Our statistician has confirmed that in her judgment, the case-control framework is the most appropriate, but we did confuse the issue with our original wording choice in the quoted passage. 

Table 2: I understand that for logistical reasons the different regions were surveyed at different times of year as were the control sites. However, it is worth some consideration of how this might influence your results. Many of the co-variates in the table will remain constant over the year, but some, e.g. % forb and %grass cover, I imagine would vary? 

Although Pinyon Jay observations were not seasonally standardized over the three study areas, collection of FIA habitat assessment data for Pinyon Jay and control locations always occurred within the latitude-defined time window allowed by the FIA protocol (which omitted the winter non-growing season). USFS chose this time window as a reasonable way to balance the need for standardized data with the reality of field work logistics. Specifically, the FIA protocol permits assessments between April and November within the latitude band that characterizes our study areas and most of the areas from where control sites were drawn. There were a handful of exceptions among the control sites only, but all Pinyon Jay use site assessments did occur within this time window. We concur that the variation in assessment time that did occur could have had some effect on measurement of forb, and to a lesser extent, grass cover. Some text has been added to make all of this clearer in the revision. 

342: Why did you chose to do the ordination separately for each region? It would be useful here to see if the different behaviour types, or control sites, cluster together regardless of region. This could inform the logistic regression.

This is an excellent point, and we frankly scratched our head trying to remember why these were originally done individually by region. In the end, we agree that a single ordination across regions is probably more informative (and certainly more consistent with the logistic regressions) and redid the analysis at the whole-project level. Perhaps because of the greater variety encompassed, NMDS was not behaving ideally in terms of finding repeat solutions, so we instead used PCA. The revision incorporates this, and the overall impressions one receives are the same as they were when looking at the regionally-specific NMDS ordinations. However, the single PCA had the advantage of creating interpretable axes, and these bolster the general interpretation of our findings very nicely, so we ended up being very happy with this alternative ordination. Some new text is devoted to describing the interpretation of PCA results and axes in biological terms. 

361: ‘Because the ratio of Pinyon Jay locations to control sites was low, the odds ratio output from logistic regression can be treated as an approximation to the resource selection function [42, 67, 68].’ It is my understanding that under certain circumstances use-availability models can approximate the resource selection function. I thought this was when the probability of use was low across all the range of co-variates, rather than when the ratio of use to available sites in the model was low?

Given the clarification we made about the case-control design above, this comment becomes moot. 

374:376: You may be right but it would be good to know why you don’t want to ‘re-use’ control sites in the different behaviour models? This seems to limit the number of control sites you can have, which several authors suggest should be maximised. Also, if I understand this correctly you end up with quite different ratios of use to control sites in the different models, specifically with very few control sites in the Foraging model. Also, all models will be dominated by control sites from Eastern Nevada, as there are more of them. An alternative approach would be to define the ratio of use to control sites you want and then select the appropriate number of control sites at random from the regions in that model.

A version of this same issue was raised by reviewer #1 and answered above in detail. In short, “reusing” control sites across the three behavior type models would have inflated type 1 error, which we sought to avoid. Upon consideration, we agree that we did not explain the rationale for our approach particularly well, and have endeavored to make changes that do a better job in this regard. We do consider the approach that we took valid given the various considerations and the inherent spatial structure of the control sites that were available to us. 

379: Logistic regression, would be good to spell out the structure of model. This is apparent from Table 4, but it would be good to clarify in the methods.

This is a good point. Some text has been added to address this comment. 

379: I am not convinced about having ‘region’ as a random effect as it only has two or three levels depending on the model. It is usually recommended that a random effect have at least six levels and ideally considerably more as it becomes difficult to estimate the model parameters below this. I suspect that is one reason why some of your models didn’t converge and why you got a lot of zero estimates for the random term in the foraging model (469:474). I would suggest you treat ‘region’ as a fixed effect or if you do pool the regions in the ordination and the control sites show considerable clustering you could consider dropping region from the model, which would give a far bigger pool of control sites to select from in each iteration.

This topic is a little beyond my (the primary author’s) expertise level, so I’ll quote our statistician’s response to this review comment, as follows: 

“I'm fine with random effects with few levels, and don’t think there’s any hard and fast rule (the Gotelli and Ellison reference already cited supports our approach to region, for example). In fact, random and fixed effects basically merge into each other at the level of complexity we're dealing with, so it's sort of a tomato-tomato thing. It is worth clarifying that the convergence issues were only and always at the permutation level, not at the model level, and they were attributable to “unreasonable splits of data” that could occur occasionally within the randomization (e.g., when the control data were split systematically along a given variable)". 

 We have added a few words in the revision to better clarify the convergence issue. We agree that the approach suggested by the reviewer could potentially strengthen the logistic regression models (though we are not surprised that the foraging model is weaker than the others given the issues mentioned in the detailed discussion above in a response of one of Reviewer # 1’s questions), but given the difference of opinion between our statistician and the reviewer about whether region is best treated as a random or fixed effect, and given that (revised) ordinations, data summaries, and existing logistic regression results are all telling a pretty consistent biological story, we would prefer to stand with what we have and get these finding out to the people who need them sooner rather than later if that is acceptable. 

379: For nesting model, would it be better to select control sites surveyed at the relevant time of year? I ask this as I see that % forb has some effect and this is likely to vary seasonally. If this is not possible due to the available sample of control sites, it is worth some discussion.

Please see the response above. All assessment data (both Pinyon Jay use sites and control sites) were collected within standardized time windows allowed by the FIA protocol, with very rare exceptions that occurred only within a handful of control sites. This is now better explained in the revision. In principle, we could have constrained the control sites further to use only those surveyed during the sub-period of the FIA data-collection window that corresponds to Pinyon Jay breeding season, but in our judgment the costs of doing this in terms of sample size would have probably not justified the improved characterization of seasonally-specific forb availability.

---

## [Editor Report · Decision Letter 1]

28 Dec 2020

Behavior-specific occurrence patterns of Pinyon Jays (Gymnorhinus cyanocephalus) in three Great Basin study areas and significance for pinyon-juniper woodland management

PONE-D-20-21672R1

Dear Dr. Boone,

We’re pleased to inform you that your manuscript has been judged scientifically suitable for publication and will be formally accepted for publication once it meets all outstanding technical requirements.

Kind regards,

Bi-Song Yue, Ph.D

Academic Editor

PLOS ONE

---

## [Editor Report · Acceptance letter]

5 Jan 2021

PONE-D-20-21672R1 

Behavior-specific occurrence patterns of Pinyon Jays (*Gymnorhinus cyanocephalus*) in three Great Basin study areas and significance for pinyon-juniper woodland management 

Dear Dr. Boone:

I'm pleased to inform you that your manuscript has been deemed suitable for publication in PLOS ONE. Congratulations! Your manuscript is now with our production department. 

Kind regards, 

on behalf of

Dr. Bi-Song Yue 

Academic Editor

PLOS ONE